# PROTECTING AGAINST SIMULTANEOUS DATA POISONING ATTACKS

**Neel Alex**[*]
University of Cambridge

**Shoaib Ahmed Siddiqui**[*]
University of Cambridge

**Amartya Sanyal**
Department of Computer Science, University of Copenhagen

**David Krueger**
Mila, University of Montreal

## ABSTRACT

Current backdoor defense methods are evaluated against a single attack at a time. This is unrealistic, as powerful machine learning systems are trained on large datasets scraped from the internet, which may be attacked multiple times by one or more attackers. We demonstrate that multiple backdoors can be simultaneously installed in a single model through parallel data poisoning attacks without substantially degrading clean accuracy. Furthermore, we show that existing backdoor defense methods do not effectively defend against multiple simultaneous attacks. Finally, we leverage insights into the nature of backdoor attacks to develop a new defense, BaDLoss (**Ba**ckdoor **D**etection via **Loss** Dynamics), that is effective in the multi-attack setting. With minimal clean accuracy degradation, BaDLoss attains an average attack success rate in the multi-attack setting of 7.98% on CIFAR-10, 10.29% on GTSRB, and 19.17% on Imagenette, compared to the average of other defenses at 63.44%, 74.83%, and 41.74% respectively. BaDLoss scales to ImageNet-1k, reducing the average attack success rate from 88.57% to 15.61%.[1]

## 1 INTRODUCTION

Many deep learning applications use large-scale datasets obtained through web scraping with minimal curation. These datasets are vulnerable to attackers, who can easily inject data that alters the behavior of models trained on these datasets. Carlini et al. (2023) demonstrated that poisoning real-world, large-scale datasets is a feasible threat due to their distributed nature, as only a few image sources need to be subverted to launch a successful attack.

Among the various data poisoning threats, the creation of model backdoors is particularly insidious. By modifying only a small number of examples in a dataset, adversaries can make a trained model sensitive to highly specific features. The adversary can then control the model's outputs by injecting these features into otherwise innocuous images (Gu et al., 2017; Liu et al., 2018b; Barni et al., 2019) – despite the model appearing benign during regular evaluation.

Almost all prior work only evaluates one backdoor attack at a time (Wang et al., 2019; Huang et al., 2023; Zhang et al., 2023). However, real-world poisoned datasets can include multiple backdoors of different types due to the presence of multiple independent attackers. Even a single attacker could deploy multiple attacks to maximize the odds of one succeeding. This setting is both more realistic and more challenging to defend against, and has seen only limited analysis (Xiang et al., 2022).

Therefore, this work evaluates the problem of defending against *multiple simultaneous backdoor attacks*. We demonstrate that a poisoned dataset can introduce multiple backdoors into a target model without substantial clean accuracy degradation. Furthermore, we show that existing defenses fail to effectively defend in this setting. As an additional contribution, we identify a common property of poisoned images in datasets: they demonstrate anomalous loss trajectories as a result of using unnatural features for classification. Consequently, we propose a defense called **BaDLoss**, meant to be robust to multiple simultaneous backdoor attacks. We demonstrate that BaDLoss outperforms other methods in the multi-attack setting, while retaining performance in the single-attack setting.

---

[*]Equal contribution. Correspondence to `sma92@cam.ac.uk`
[1]We open-source our code to aid replication and further study, available on GitHub: `https://github.com/shoaibahmed/badloss/`

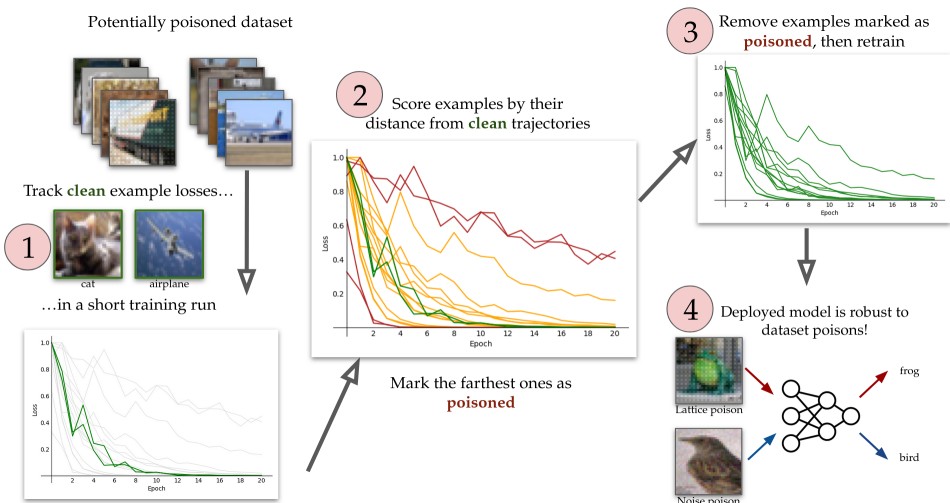

Figure 1: **BaDLoss Overview.** (1) The defender tracks bona fide clean examples in the training set in a short training run. (2) Every example gets an anomaly score based on its average distance from the bona fide clean examples. The farthest examples are marked as potential backdoors. (3) The defender retrains the model, excluding any examples identified as anomalous. (4) The defender deploys the robust model.

## 2 RELATED WORK

Since our work is concerned with backdoor detection and prevention, we briefly discuss both attacks as well as defenses presented in the past. Most work on backdoors, including ours, focuses on vision models. Several recent works explore the poisoning of language models (Wan et al., 2023; Hubinger et al., 2024; Xu et al., 2024), which are beyond the scope of this work.

### 2.1 BACKDOOR ATTACKS

While attacks exist where the adversary provides a secretly backdoored model to the victim (Nguyen & Tran, 2020; Tan & Shokri, 2020), we focus on **data poisoning**, where the attacker manipulates a subset of the training data to later control the model by injecting a feature of their choice.

Early data poisoning attacks modified both training images and corresponding labels: adding a small trigger (Gu et al., 2017), overlaying an image (Chen et al., 2017) or pattern (Liao et al., 2018), followed by altering the label to the **target class**. The trained model thus associates the injected feature with the target class. Later attacks invisibly warped images (Nguyen & Tran, 2021) or overlaid image-specific invisible perturbations (Li et al., 2021c).

Another class of methods, **clean-label** attacks, require no alteration to the image's label. Clean-label attacks alter training images with features that the learning algorithm preferentially detects (Turner et al., 2019; Barni et al., 2019) without changing any image classes. More sophisticated strategies used realistic reflection overlays (Liu et al., 2020) or random-noise patterns (Souri et al., 2022; Zeng et al., 2022b) to implant the backdoor while minimizing the risk of detection.

### 2.2 BACKDOOR DEFENSES

There exist many defense mechanisms to mitigate backdoor attacks. One approach, known as trigger reverse engineering, assumes that a mask with very few pixels triggers the model to misbehave. Neural Cleanse (Wang et al., 2019) and variants (Guo et al., 2019; Tao et al., 2022; Dong et al., 2021; Wang et al., 2020) learn trigger masks from a trained model. Other techniques use activation patching (Liu et al., 2019) or generative modeling (Qiao et al., 2019) to identify the trigger. Subsequently, neurons associated with the trigger can be pruned, or other cleaning methods can be applied.

Backdoors can also be detected by examining model activations. Activation Clustering and Spectral Signatures (Tran et al., 2018) assumed that clean and backdoored examples appear as linearly separable clusters in the model's activation space, allowing the defender to remove identified backdoors. Further work examined specific properties of activation space and identified neurons that highly

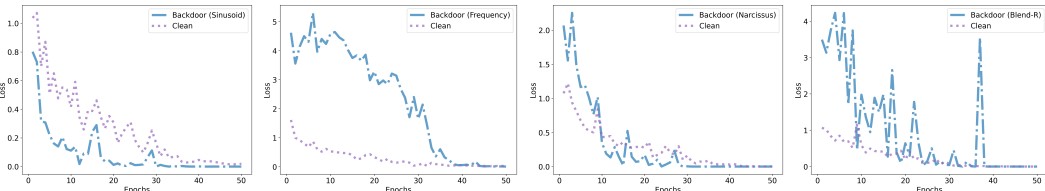

Figure 2: Selected average clean trajectories compared to average attack trajectories in CIFAR-10, single-attack. **Left to right:** Sinusoid, frequency, narcissus, random-blend attack. All backdoor attacks exhibit distinct learning dynamics from clean examples. However, these dynamics vary between faster than clean examples, slower than clean examples, or a combination, making the assumptions of previous methods (Li et al., 2021a; Khaddaj et al., 2023; Hayase et al., 2021) inappropriate for defending against general poisoning attacks.

influence model behavior on backdoored examples (Zheng et al., 2022; Wu & Wang, 2021; Xu et al., 2021). These neurons can then be pruned (Liu et al., 2018a), or repaired via careful finetuning of the model (Chen et al., 2019; Li et al., 2021b; Zeng et al., 2022a).

As removing backdoors from a trained model is challenging (Goel et al., 2022), many defenses indirectly prevent the backdoor from being learned at all (Hong et al., 2020; Borgnia et al., 2021; Huang et al., 2022; Wang et al., 2022). One method, Anti-Backdoor Learning (Li et al., 2021a) identifies poisoned examples by their low loss at a given epoch, then *maximizes* the loss of identified examples, making the model ignore poisoned features. Subsequently, ABL has been combined with other more powerful detection techniques (Chen et al., 2022; Huang et al., 2023).

While BaDLoss resembles ABL in that we use loss as a signal for whether an example is poisoned, ABL is a special case of BaDLoss's formulation, which permits more general identification schemes. In particular: (i) BaDLoss considers the entire trajectory of losses at every epoch, and (ii) BaDLoss identifies both unusually *low-loss* as well as *high-loss* examples as poisoned, representing different training dynamics explicitly explored in past work. We find that this model is more robust, as BaDLoss outperforms ABL in our experiments. We further explore these differences in Appendix B.

### 2.3 STUDYING TRAINING DYNAMICS

Differences in training dynamics between different classes of examples have been previously studied (Arpit et al., 2017), and these dynamics have been harnessed for various objectives (Kaplun et al., 2022; Liu et al., 2022; Rabanser et al., 2022). BaDLoss draws inspiration from the Metadata Archaeology via Probe Dynamics (MAP-D) technique (Siddiqui et al., 2022), which examines per-example loss trajectories in a training set to identify different types of examples.

## 3 MULTIPLE SIMULTANEOUS ATTACKS

To the best of the authors' knowledge, only one prior work (Xiang et al., 2022) evaluates on multiple backdoor attacks simultaneously. This work is limited, and does not examine the practical feasibility of actually deploying multiple simultaneous attacks. Therefore, we will begin with a demonstration that this setting is feasible in practice without substantial clean-accuracy degradation.

We compare against a wide variety of classic and state-of-the-art backdoor attacks and defenses in the literature. Following recent work (Wang et al., 2022; Kiourti et al., 2021; Li et al., 2021b; Do et al., 2023), we use the standard computer vision datasets: CIFAR-10 (Krizhevsky, 2009), GTSRB (Houben et al., 2013), and Imagenette (Howard, 2019), which capture variance both in image size (32px square for CIFAR-10, versus variable image sizes between 25px and 243px in GTSRB, and between 27px and 4368px in Imagenette) as well as class distribution (CIFAR-10 and Imagenette have 10 balanced classes, while GTSRB has 43 imbalanced classes). We use CIFAR-10 as a development set to tune defenses and apply the tuned methods directly to GTSRB and Imagenette *without any further tuning* for an accurate assessment of the methods' performance in novel settings.

### 3.1 THREAT MODEL

In our threat model, we assume that the attacker(s) have control over a portion of the training dataset (called the poisoning ratio $p$), in which they can modify both images and their labels. The defender

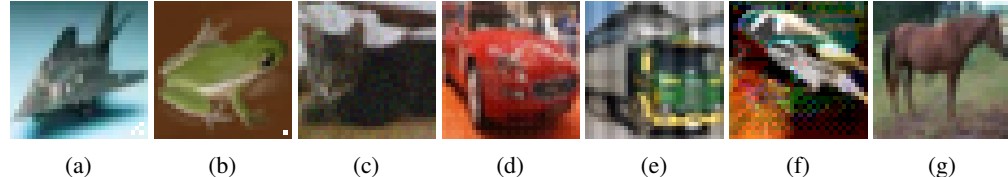

|   (a)   |   (b)   |   (c)   |   (d)   |   (e)   |   (f)   |   (g)   |

Figure 3: Complete list of attacks considered in this work. **(a)** checkerboard pattern trigger (Patch) Gu et al. (2017), **(b)** single pixel trigger (Single-Pix) Gu et al. (2017), **(c)** random noise blending attack (Blend-R) Chen et al. (2017), **(d)** dimple pattern blending attack (Blend-P) (Liao et al., 2018), **(e)** sinusoid pattern blending attack (Sinusoid) (Barni et al., 2019), **(f)** optimized-trigger attack (Narcissus) (Zeng et al., 2022b), and **(g)** frequency-domain attack (Frequency) (Wang et al., 2021)

controls all details of the training process (e.g. algorithms, architecture, hyperparameters, etc.; the attacker has no knowledge of these details) and receives a labeled dataset from an unvalidated external source. This reflects the real-world setting where a victim trains a model using trusted, internal code, but relies on externally-sourced datasets that cannot practically be manually inspected and validated. Therefore, the defender must deploy a training method that minimizes the influence of poisoned examples. These assumptions align with the threat models commonly considered in prior works (Carlini et al., 2023; Chen et al., 2018; Li et al., 2021a; Do et al., 2023).

Additionally, we assume that the defender has access to a small set of guaranteed clean examples (250 examples in our case). Many defense methods (Liu et al., 2018a; Wang et al., 2019; Gao et al., 2020; Chou et al., 2020; Kiourti et al., 2021; Li et al., 2021b; Zeng et al., 2022c; Wang et al., 2022; Huang et al., 2022) assume access to bona fide clean examples, as trusted human labor could manually generate or filter a subset of the dataset.

Our framework focuses on retraining the model after filtering the detected backdoor examples from the dataset. Subsequently, we evaluate the retrained model's performance on the original task and the success rate of any backdoor attacks. This is consistent with defense evaluations considered in prior works (Li et al., 2021a; Chen et al., 2018; Wang et al., 2019). We define clean accuracy as the retrained model's performance on the test set. Attack success rate (ASR) is evaluated on the full test set *excluding* the target class, with the backdoor injected into every example to verify that the attack successfully changes the model's predictions. Full training details are available in Appendix H.

### 3.1.1 MULTI-ATTACK ASSUMPTIONS

We assume that multiple attacks can *not* target the same image, i.e. a given image has at most one trigger. We explore this assumption further in Appendix F. However, multiple attacks can target the same label, which enables evaluation of certain unusual attack/defense interactions.

### 3.2 ATTACKS CONSIDERED

The attacks considered in our work are visualized in Figure 3. All attacks perturb the target label in addition to the input image, except for the sinusoid and narcissus attacks, which are clean-label attack. In particular, we evaluate against the following attacks:

- **Patch**: Adds a small patch in the corner of attacked images (Gu et al., 2017).
- **Single-Pixel**: Adds a single pixel in the corner of attacked images (Gu et al., 2017).
- **Blend-Random**: Blends a random noise pattern with attacked images (Chen et al., 2017).
- **Blend-Pattern**: Blends a dimple pattern with attacked images (Liao et al., 2018).
- **Sinusoid**: Adds sinusoidal stripes to attacked images of a single class (Barni et al., 2019).
- **Narcissus**: Adds a learned trigger to a very small number of images (Zeng et al., 2022b).
- **Frequency**: Adds peaks in attacked image's discrete cosine transform (Wang et al., 2021).

We only evaluated the narcissus attack on CIFAR-10, as the optimized trigger is challenging to generate on variable-size images. Additionally, we did not evaluate the single-pixel attack on Imagenette, as very large image sizes make learning a one-pixel trigger challenging.

In the multi-attack setting, all attacks are present at their full poisoning ratio. Full attack specifications are available in Appendix I.1.

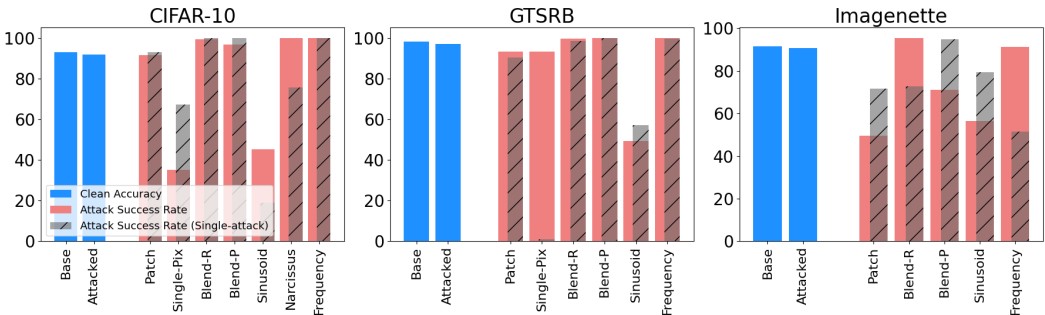

Figure 4: **Evaluation of Multi-attack Setting.** Clean accuracy (blue) should be high, attack success rate (red) should be low. When multiple attacks are simultaneously deployed, clean accuracy is barely degraded. The model simultaneously learns all attacks. We plot the attack success rate that each attack achieves when deployed on its own to show positive and negative interference between concurrent attacks.

## 3.3 Efficacy of Multiple Attacks

Our results are visualized in Figure 4 (numerical results in Table 2). In all datasets, the presence of multiple attacks degrades clean accuracy by less than 1%. Additionally, all attacks are learned simultaneously to a high degree. Notably, some attacks substantially change in performance between the single-attack and multi-attack setting – for instance, the single-pixel attack achieves over 90% success rate in GTSRB's multi-attack setting despite failing when deployed as a single attack. This positive interaction between co-located attacks has previously been observed by Schneider et al. (2024). This indicates that even attack-development research should consider the multi-attack setting, as concurrent attacks can dramatically impact an attack's viability in realistic threat models.

These results validate the importance of the multi-attack setting. If one or more attackers can manipulate small portions of an internet-scraped dataset to make a target model react to various backdoor triggers, all while maintaining high performance on clean data (i.e., without raising suspicion from the defender), this scenario becomes critically important. We must therefore examine whether current defenses can effectively adapt to this new setting.

## 3.4 Defenses Considered

In this work, we evaluate the following defenses[2]:

- **Neural Cleanse** (Wang et al., 2019) Neural Cleanse learns a minimum-magnitude mask per class to reclassify every image in the dataset as that class. Masks with substantially-below-median magnitude are considered backdoor triggers.

- **Activation Clustering** (Chen et al., 2018) Activation Clustering determines that a backdoor is present if the last layer's activations for a given class can be clustered into two classes, then removes the smaller cluster for retraining.

- **Spectral Signatures** (Tran et al., 2018) Spectral signatures uses a singular value decomposition of last layer activations per class, and removes the 15% of data with the highest value in the first singular dimension for retraining.

- **Frequency Analysis** (Zeng et al., 2022c) Frequency analysis identifies poisoned examples by building a classifier on the discrete cosine transforms of synthetic images with a fixed set of hardcoded backdoor-like features.

- **Anti-Backdoor Learning** (Li et al., 2021a) Anti-backdoor learning (ABL) marks examples with low loss after a few epochs of training as backdoored. The model is then subsequently finetuned using gradient descent on clean examples, and gradient ascent on backdoored examples.

- **Cognitive Distillation** (Huang et al., 2023) Cognitive distillation learns a mask for each example that yields the same activations, treating examples with particularly low magnitude as backdoors, then uses ABL's unlearning method.

---

[2]We do not compare against the Expected Transferability detection method proposed by Xiang et al. (2022), despite this method being evaluated on the multi-attack setting, as this method only detects the presence of a backdoor rather than removing the backdoor. Consequently, it cannot be evaluated with standard metrics.

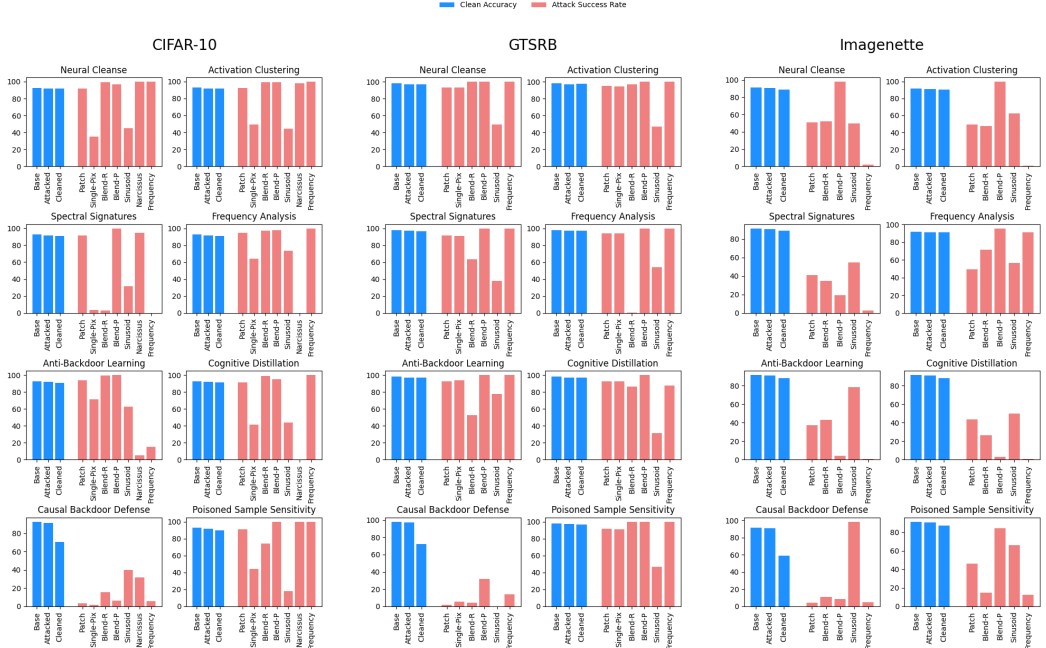

Figure 5: **Defense Performance in the Multi-attack Setting.** All evaluated defenses exhibit failures across datasets when evaluated in the multi-attack setting. CBD (Zhang et al., 2023) can achieve low attack success rates, but causes extreme clean accuracy degradation. On CIFAR-10, Spectral Signatures provides the best overall defense without substantial clean accuracy degradation, but still underperforms BaDLoss (see Figure 6). In Imagenette, attacks are weaker yet every defense still lets one or more attacks succeed.

- **Causal Backdoor Defense** (Zhang et al., 2023) Causality-inspired backdoor defense pretrains a weak classifier that is expected to learn backdoors more than clean images, then trains a strong classifier to be causally decoupled from the weak one, removing the backdoor's influence.
- **Poisoned Sample Sensitivity** (Chen et al., 2022) Poisoned sample sensitivity uses the sensitivity towards transformations to identify poisoned samples, then either filters those samples out, or uses an unlearning procedure similar to ABL.

Full defense specifications are available in Appendix I.2.

## 3.5 EXISTING DEFENSE PERFORMANCE

Our results are visualized in Figure 5 (numerical results in Table 2). Except for the causal backdoor defense, which shows substantial clean accuracy degradation, all defenses in the multi-attack setting fail against a variety of attacks. Even the *best* performing defenses, Spectral Signatures on CIFAR-10 and Cognitive Distillation on Imagenette, have three attacks completely bypass the defense.

Likely due to making assumptions incompatible with the complexity of the multi-attack setting, current backdoor defenses do not effectively defend against these standard attacks when deployed simultaneously. To make progress in the multi-attack setting, new defenses must be developed.

## 4 DEVELOPING A NEW DEFENSE

To defend in the multi-attack setting, we first note that many assumptions made by existing defenses, while accurate in a single-attack setting, no longer hold in the presence of multiple simultaneous attacks. For instance, neural cleanse assumes that the attacked class will have an anomalously low mask magnitude relative to the other classes, but if sufficiently many classes are attacked, the median mask magnitude may be low enough such that no classes look anomalous. For further analysis of why existing defenses might be failing in the multi-attack setting, refer to Appendix E.

Several methods exist that analyze the loss of training examples to determine whether such examples are installing a backdoor. Anti-Backdoor Learning (Li et al., 2021a) assumes that poisoned examples

achieve a lower loss more quickly than normal examples, and Khaddaj et al. (2023) similarly claim that backdoor images have the *strongest* features and are thus learned more easily. Other works (Hayase et al., 2021) treat backdoors as anomalous examples which conflict with natural features, and are therefore harder to learn. These works suggest that because backdoor images contain unnatural features or incorrect labels, these examples demonstrate unusual training dynamics.

These findings demonstrate that backdoors can be detected by analyzing the training dynamics of individual examples, regardless of whether they exhibit faster or slower learning compared to clean data. The pattern may even be more complex; for instance, on CIFAR-10, the Narcissus attack examples have higher loss until epoch 10 and lower loss thereafter (see Figure 2). As a defender cannot know which backdoor methods will be deployed, a defense must be effective against backdoors that induce *any* unusual training dynamics.

### 4.1 BaDLoss: Backdoor Detection via Loss Dynamics

We introduce **BaDLoss** *(Backdoor Detection via Loss Dynamics)*, a method of identifying poisoned examples based on their anomalous training dynamics. BaDLoss is visualized in Figure 1. Inspired by MAP-D (Siddiqui et al., 2022) and unlike previous defense methods such as Anti-Backdoor Learning, BaDLoss considers the entire *trajectory* of losses achieved on different examples, enabling more precise detection of backdoored examples (see Appendix B for further discussion). It leverages bona fide clean trajectories to define an anomaly score for each training example.

BaDLoss leverages a dataset $\mathcal{D}_\mathbf{c}$ comprising a small number (e.g. 250) of *bona fide* clean examples. These examples are included in the training set $\mathcal{D}$ and the loss $\ell$ of every training example $(x_i, y_i)$ is tracked at each epoch as the model is partially trained[3], creating **loss trajectories**, $\mathbf{s}_i$:[4]

$$\mathbf{s}_i^t := (\ell(x_i, y_i; \theta_1), \ell(x_i, y_i; \theta_2), ..., \ell(x_i, y_i; \theta_t) \mid (x_i, y_i) \in \mathcal{D}) \tag{1}$$

where $\mathbf{s}_i$ represents the loss trajectory for example $i^{th}$ example: $(x_i, y_i)$, and $\theta_t$ represents the weights of the network at iteration $t$. Treating each example's loss trajectory as a vector, we calculate a score for each example as the log of the mean $\ell_2$ distance between the target trajectory and the nearest bona fide clean examples.

$$p(b \mid \mathbf{s}_i) \propto \sum_{\mathbf{g} \, \in \, \text{NN}(\mathbf{s}_i, \mathcal{D}_\mathbf{c}, k)} \ln \left( ||\mathbf{s}_i^t - \mathbf{g}||_2 + \epsilon \right) \tag{2}$$

where $p(b \mid \mathbf{s}_i)$ represents the probability of the example being backdoored given the full training trajectory, and $\mathcal{D}_\mathbf{c}$ represents the set of clean bona fide examples. $\epsilon = 1\mathrm{e}{-8}$ for stability if the distance is very small. We identify a fixed percentage of examples – those with the highest average distance from the clean trajectories – as backdoored. We then retrain the model on the original dataset with the identified backdoored examples removed.

Several hyper-parameters must be selected: $n_{clean}$, the number of bona fide clean training examples; $k$, the number of clean nearest neighbors examples to score with; $r_{epoch}$, the fraction of clean-training epochs to use for detection; and $r_{reject}$, the fraction of samples to reject.

## 5 Results

### 5.1 Multi-Attack Result

Throughout our experiments, we set $(n_{clean}, k, r_{epoch}, r_{reject}) = (250, 50, 0.3, 0.4)$: we always track 250 bona fide clean examples and use the 50 nearest ones to calculate an example's anomaly score. We pretrain for 30% of the epochs that are used to train the final model (so, 30 epochs in CIFAR-10 and GTSRB, which are trained for 100 epochs, and 75 epochs on Imagenette, which is trained for 250 epochs), as this is sufficient for backdoored examples to produce different loss trajectories without becoming too computationally expensive.

---

[3]We used the same training method here as when the final model is trained, but Qi et al. (2023) note that the defender may control their pre-training procedure more generally. Natural options would be to increase the batch size and adjust the model architecture in pre-training to produce more stable and separable loss trajectories.

[4]In the multi-attack setting, the average training loss often undergoes large spikes, visible in Figure 9. To mitigate this, we do not track any epoch whose average training loss is higher than twice the moving average of the last three (non-rejected) epochs.

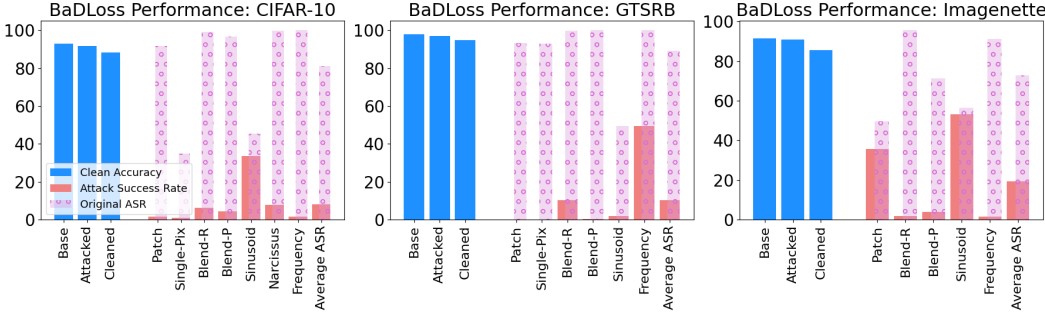

Figure 6: **Evaluation of BaDLoss in the Multi-attack setting.** BaDLoss achieves very low attack success rate across all except one attack on CIFAR-10 and GTSRB, and except two attacks on Imagenette. In all cases, it substantially lowers the average attack success rate. This figure should be considered in conjunction with Figure 5, which highlights the failures of existing defenses.

The rejection threshold $r_{reject}$ most impacts the trade-off between improving defense quality and degrading clean performance. We visualize the impact of this threshold on clean accuracy as well as the average Attack Success Rate (ASR) in Figure 7. The figure highlights that $r_{reject} = 0.4$ is an ideal point to minimize clean accuracy degradation while preventing most attacks from succeeding. Therefore, we always reject the 40% of training examples with the greatest distance to the clean trajectories in all subsequent experiments. Further ablation results on the filtering threshold are presented in Appendix J.1, including its performance on other datasets.

The overall fraction of the dataset which is poisoned is approximately 8% on CIFAR-10, 10% on GTSRB, and 9% on Imagenette. The difference is due to required adjustment of the attack strength to ensure that the attacks are correctly learned for each dataset.

BaDLoss' defense results are visualized in Figure 6 (numerical results available in Table 2). BaDLoss is the first defense to credibly defend in the multi-attack setting, substantially reducing ASR on every attack considered. In all three datasets, BaDLoss defends against nearly every attack with only minor clean accuracy degradation.

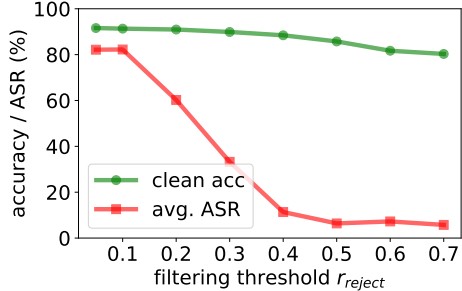

Figure 7: Impact of filtering threshold $r_{reject}$ on the clean accuracy as well as the avg. ASR on CIFAR-10, multi-attack setting.

## 5.2 IMAGENET EVALUATION

To determine whether BaDLoss effectively scales to large and complex datasets, we evaluate BaDLoss's performance on ImageNet-1k (Deng et al., 2009). The dataset is comprised of 1.2 million images and 1,000 classes, exceeding our main datasets both in terms of size and complexity. For computational convenience, we use $n_{epochs} = 50$ and use a ResNet-18 architecture instead of ResNet-50. We use optimized settings from CIFAR-10 and the same attack set as Imagenette. The overall poisoning ratio is 0.28% on ImageNet – substantially lower than other datasets. The results are presented in Figure 8. First, we observe that multiple simultaneous attacks are effectively learned on ImageNet despite their much lower poisoning ratios, demonstrating again that the multi-attack setting represents a realistic threat model. Second, we observe that BaDLoss effectively scales to this larger setting by maintaining adequate clean accuracy while reducing ASR on every single attack type. Experimental details are available in Section H.

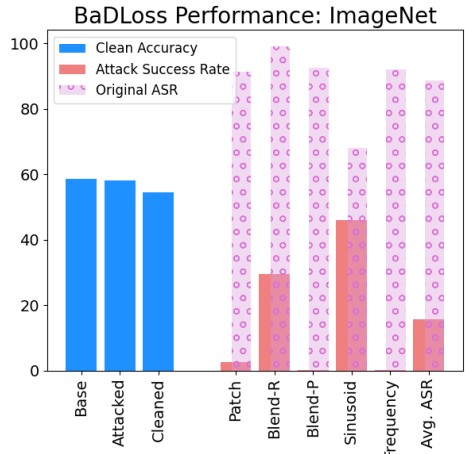

Figure 8: Multiple attacks are effectively learned in ImageNet-1k without substantial clean-accuracy degradation, and BaDLoss effectively defends against those attacks. Numerical results in Table 3.

|  |  | Clean Accuracy | | | | | | | | Attack Success Rate | | | | | | | |
|---|---|---|---|---|---|---|---|---|---|---|---|---|---|---|---|---|---|
|  |  | Patch | Single-Pix | Blend-R | Blend-P | Sinusoid | Narcissus | Frequency | Avg. CA | Patch | Single-Pix | Blend-R | Blend-P | Sinusoid | Narcissus | Frequency | Avg. ASR |
| **CIFAR-10** | No Defense | 92.86 | 92.11 | 92.71 | 92.40 | 92.89 | 87.57 | 92.67 | 91.89 | 92.93 | 67.12 | 99.88 | 99.96 | 18.90 | 75.60 | 99.97 | 79.20 |
|  | Neural Cleanse | 92.29 | 92.11 | 92.71 | 92.84 | 92.56 | 93.07 | 92.45 | **92.58** | 91.91 | 67.12 | 99.99 | 2.48 | 39.89 | 87.06 | **0.63** | 55.58 |
|  | Activation Clustering | 90.55 | 91.96 | 92.92 | 92.76 | 92.85 | 92.07 | 92.21 | 92.19 | 86.89 | 54.57 | 99.91 | 99.87 | 30.86 | 91.03 | **0.63** | 66.25 |
|  | Spectral Signatures | 91.64 | 90.68 | 91.50 | 91.70 | 90.68 | 90.72 | 90.98 | 91.13 | 88.62 | 1.32 | 1.82 | **1.87** | 23.46 | 33.89 | 0.86 | 21.69 |
|  | Frequency Analysis | 31.66 | 32.52 | 36.56 | 33.89 | 21.14 | 44.79 | 32.14 | 33.24 | 39.78 | 12.60 | 3.83 | 5.20 | **0.50** | 2.36 | 5.02 | 9.90 |
|  | Anti-Backdoor Learning | 90.33 | 90.95 | 91.83 | 91.29 | 91.33 | 90.73 | 91.49 | 91.14 | 92.36 | 57.91 | 99.90 | 99.87 | 39.07 | 81.59 | 98.68 | 81.34 |
|  | Cognitive Distillation | 91.48 | 91.87 | 92.29 | 91.47 | 91.73 | 90.33 | 91.81 | 91.57 | 91.01 | 49.59 | 84.99 | 1.96 | 41.42 | 83.99 | 0.83 | 50.54 |
|  | Causal Backdoor Defense | 73.05 | 75.42 | 77.02 | 71.79 | 61.08 | 56.85 | 76.19 | 70.20 | 2.92 | 13.79 | 14.21 | 7.64 | 3.12 | 23.08 | 2.97 | 9.68 |
|  | Poisoned Sample Sensitivity | 92.69 | 92.02 | 92.36 | 92.23 | 92.46 | 91.76 | 91.10 | 92.09 | 93.42 | 63.83 | 99.98 | 99.88 | 45.26 | 71.67 | 99.92 | 81.99 |
|  | BaDLoss | 85.49 | 84.78 | 85.22 | 83.98 | 84.72 | 84.91 | 86.25 | 85.05 | **1.02** | **0.92** | **1.66** | 3.07 | 19.92 | **0.72** | 0.82 | **4.02** |
| **GTSRB** | No Defense | 97.02 | 95.80 | 97.93 | 98.22 | 97.89 | - | 98.94 | 97.63 | 90.37 | 1.00 | 98.50 | 99.96 | 56.95 | - | 99.67 | 74.41 |
|  | Neural Cleanse | 94.83 | 95.80 | 96.82 | 97.82 | 97.89 | - | 98.21 | 96.90 | 0.36 | 1.00 | **0.00** | **0.00** | 56.95 | - | **0.00** | 9.72 |
|  | Activation Clustering | 97.16 | 95.13 | 98.17 | 97.69 | 98.16 | - | 97.51 | **97.30** | **0.16** | 1.03 | 97.26 | 100.00 | 59.92 | - | 98.64 | 59.50 |
|  | Spectral Signatures | 96.75 | 96.06 | 97.32 | 97.69 | 97.47 | - | 97.70 | 97.17 | 0.55 | 0.80 | 97.04 | 100.00 | 6.84 | - | 100.00 | 50.87 |
|  | Frequency Analysis | 95.27 | 95.50 | 97.95 | 98.04 | 97.93 | - | 97.89 | 97.10 | 1.36 | 0.74 | 98.58 | 99.82 | 66.42 | - | 100.00 | 61.15 |
|  | Anti-Backdoor Learning | 93.56 | 97.77 | 95.55 | 97.28 | 96.41 | - | 97.13 | 96.28 | 1.97 | 2.37 | 98.62 | 99.99 | 61.11 | - | 8.62 | 45.45 |
|  | Cognitive Distillation | 97.02 | 96.31 | 97.88 | 97.51 | 97.47 | - | 97.54 | 97.29 | 74.71 | 0.53 | 98.53 | 99.66 | 45.49 | - | 99.51 | 69.74 |
|  | Causal Backdoor Defense | 43.64 | 31.27 | 33.67 | 27.74 | 7.38 | - | 20.63 | 37.39 | 13.55 | 47.71 | 0.96 | 1.15 | **0.04** | - | 10.03 | 12.24 |
|  | Poisoned Sample Sensitivity | 97.14 | 87.47 | 95.87 | 98.12 | 96.10 | - | 97.85 | 95.43 | 91.24 | 10.30 | 97.01 | 100.00 | 55.48 | - | 99.71 | 75.62 |
|  | BaDLoss | 94.34 | 96.19 | 94.82 | 93.94 | 93.90 | - | 94.71 | 94.65 | 0.31 | **0.04** | 94.50 |  | 60.70 | - | 33.83 | 32.70 |
| **Imagenette** | No Defense | 90.01 | - | 91.13 | 91.16 | 91.34 | - | 91.46 | 91.02 | 71.75 | - | 72.66 | 94.90 | 79.35 | - | 51.53 | 74.04 |
|  | Neural Cleanse | 89.73 | - | 91.11 | 91.57 | 91.67 | - | 91.11 | 91.04 | 55.70 | - | 66.51 | 84.15 | 52.35 | - | 70.65 | 65.87 |
|  | Activation Clustering | 90.37 | - | 91.90 | 91.44 | 91.44 | - | 91.90 | **91.41** | 51.11 | - | **0.48** | 90.50 | 90.73 | - | 1.79 | 46.92 |
|  | Spectral Signatures | 89.55 | - | 90.75 | 90.22 | 91.11 | - | 89.81 | 90.29 | 54.79 | - | 45.12 | 45.04 | 73.82 | - | 1.30 | 44.01 |
|  | Frequency Analysis | 90.01 | - | 91.13 | 91.16 | 91.34 | - | 91.46 | 91.02 | 71.75 | - | 72.66 | 94.90 | 79.35 | - | 51.53 | 74.04 |
|  | Anti-Backdoor Learning | 88.89 | - | 90.29 | 91.03 | 90.90 | - | 90.24 | 90.27 | 49.04 | - | 58.59 | 94.84 | 83.30 | - | 69.09 | 70.97 |
|  | Cognitive Distillation | 88.66 | - | 89.94 | 89.89 | 90.22 | - | 89.76 | 89.69 | 63.44 | - | 8.79 | 85.05 | 61.51 | - | 70.73 | 57.904 |
|  | Causal Backdoor Defense | 70.78 | - | 67.03 | 68.38 | 63.46 | - | 68.76 | 67.68 | **3.91** | - | 20.14 | 5.73 | 48.89 | - | 5.33 | 16.80 |
|  | Poisoned Sample Sensitivity | 88.03 | - | 88.99 | 90.55 | 85.38 | - | 84.54 | 87.50 | 66.22 | - | 69.09 | 96.99 | 80.32 | - | 44.70 | 71.46 |
|  | BaDLoss | 85.45 | - | 86.11 | 86.60 | 86.39 | - | 86.50 | 86.21 | 29.15 | - | 3.66 | **1.02** | 30.60 | - | **0.82** | 13.05 |

Table 1: **Single-attack setting: Clean accuracy and attack success rate after retraining on CIFAR-10 and GTSRB.** BaDLoss is highly effective at removing detected backdoor instances, usually minimizing the backdoor's efficacy without significant degradation in model utility.

## 5.3 SINGLE-ATTACK RESULTS

Other methods have primarily been evaluated in a single-attack setting. In order to adequately compare BaDLoss against other methods, we also evaluate its performance in the single-attack setting. Our results are visualized in Table 1. BaDLoss presents a very strong defense. On CIFAR-10, BaDLoss attains near-zero attack sensitivity, on GTSRB, it performs comparably well to other defense methods, and on Imagenette, BaDLoss achieves the best overall defense while remaining competitive in clean accuracy. While the high removal fraction $r_{reject} = 0.4$ is more suitable to the multi-attack setting with its higher overall poisoning ratio, here it degrades clean accuracy relative to other methods (though a practitioner could tune it according to the expected poisoning ratio). Nonetheless, these results show that BaDLoss is effective in the single-attack setting as well.

## 6 DISCUSSION

### 6.1 PRACTICAL USE

BaDLoss causes clean accuracy degradation due to the large removal fraction needed to reduce susceptibility against all attacks. In practice, a defender can set a target clean accuracy and adjust example removal to minimize the attack-susceptibility of the model while maintaining acceptable performance. However, given that multiple simultaneous attacks appear viable in all studied domains, if backdoor attacks are considered a potential threat, we believe that a defense robust against multiple simultaneous attacks should be used.

BaDLoss's identification method could also be used with alternate backdoor removal methods, such as Anti-Backdoor Learning's gradient ascent (Li et al., 2021a), or other unlearning methods that rely on an identification module (Chen et al., 2022; Huang et al., 2022). Future work could compare cleansing methods' effectiveness given fixed identification rates, which could lead to stronger defenses by combining the best detection and cleansing methods.

### 6.2 DIFFERENCES BETWEEN DATASETS

Empirically, the multi-attack setting appears to work effectively in all datasets we tested, allowing multiple attacks to be learned with minimal clean-accuracy degradation. Across datasets, BaDLoss appears robust to variance in number of classes and image complexity. However, BaDLoss struggles

more on smaller datasets – for instance, Imagenette has only ∼9,500 train examples, is the only dataset where BaDLoss fails to defend against the Patch attack. We hypothesize that this is due to the smaller dataset size inducing more variance in the loss trajectories. Consequently, BaDLoss may not be suitable for low-data domains.

## 6.3 IMPACT OF THE POISONING RATIO

Poisoning ratios significantly impact BaDLoss's effectiveness by altering training dynamics. For example, CIFAR-10's random blending attack becomes easier to learn at higher ratios despite being harder at lower ones. Since BaDLoss detects deviations from bona fide example dynamics, it can handle both high poisoning ratios, which are learned unusually quickly and low poisoning ratios, which are learned unusually slowly. However, BaDLoss may struggle if poisoning ratios are chosen to match backdoored examples' training dynamics to those of the clean examples.

## 6.4 LIMITATIONS

We highlight some of the major limitations of our work.

**Training instabilities.** In the presence of multiple backdoor attacks, we observe instabilities in the loss trajectories in the form of abrupt loss spikes. While these instabilities settle eventually, they add significant noise to our distance computation. Despite filtering noisy epochs, this instability still harms BaDLoss's performance. The defender has considerable leeway to optimize the pretraining process for backdoor detection (Qi et al., 2023), and maximizing loss stability could significantly improve the performance of BaDLoss.

**Impact of example removal.** Our defense methodology removes examples from the training set. This is a common practice in past work, but removal can have outsized negative impacts on the long-tail of minority classes (Feldman & Zhang, 2020; Liu et al., 2021; Sanyal et al., 2022). Specifically, BaDLoss may more likely mark minority-class data as anomalous, as the reduced amount of training data may render trajectories unstable. Consequently, BaDLoss may exacerbate existing weaknesses on minority-class data. Depending on the circumstances, this weakness may be partially ameliorated by choosing bona fide clean examples such that there are at least a fixed number from each class. For further analysis, see Appendix G.

**Counter-attacks.** Generally, our assumption that attacks demonstrate anomalous loss dynamics is more robust than other models of attacks implied by other defenses (see Appendix E). However, an informed attacker can still exploit the defense mechanism, e.g. by carefully selecting injected features, poisoning ratios, and other attack parameters to ensure that their attack produces training dynamics that closely match the bona fide clean training examples. For instance, trained attack images (such as the sleeper agent attack (Souri et al., 2022)) could add a regularization term in training to ensure that their losses mimic clean examples on a proxy model. However, this adaptive attack would require multiple rounds of a complex training procedure, potentially making it infeasible for practical purposes. More details regarding adaptive attacks are presented in Appendix D.

## 7 CONCLUSION

Our work makes two central contributions. First, we examine the problem of defending against **multiple simultaneous data poisoning attacks**. We argue this is a more realistic threat model for machine learning systems trained on large datasets comprised of internet data, and show that existing defenses fail in this setting. Our second contribution is **BaDLoss**, a novel backdoor detection method that successfully defends against multiple simultaneous poisoning attacks with minimal degradation in clean accuracy. BaDLoss works by: (i) comparing the loss trajectories of individual training examples to those of a small set of known-to-be-clean examples, (ii) filtering out those examples with anomalous trajectories, and (iii) retraining from scratch on the filtered training set. Our experiments demonstrate the effectiveness of BaDLoss in both single-attack and multi-attack settings. We hope our work will inspire further research on simultaneous data poisoning attacks where we believe BaDLoss provides a strong baseline for comparison.

## 8 ACKNOWLEDGMENTS

We would like to thank Mantas Mazieka and Nicolas Papernot for their helpful guidance and feedback, as well as anonymous reviewers who provided abundant useful feedback and references.

This research was supported by the Center for AI Safety Compute Cluster. Any opinions, findings, and conclusions or recommendations expressed in this material are those of the author(s) and do not necessarily reflect the views of the sponsors.

## 9 CONTRIBUTION STATEMENT

N.A. and S.S. proposed the initial idea for the BaDLoss defense. S.S. and D.K. proposed the initial idea for the multi-attack setting. S.S. performed initial experiments demonstrating BaDLoss's viability. All authors brainstormed ways to improve the BaDLoss method. N.A. and S.S. implemented the attacks. N.A. implemented all defenses. N.A. tuned experimental parameters and performed all experiments in the final manuscript. N.A. created all figures in the final manuscript, except S.S. created Figure 7. N.A. wrote all text in the final manuscript, except S.S. wrote the first paragraph of Section 1, Section 2.3, and the middle paragraphs of Section 4.1, and D.K. wrote the final paragraph of Section 2.2. S.S. performed several comprehensive editing rounds to the manuscript, though all authors contributed edits. D.K. supervised the project.

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

## A  BADLOSS PSEUDOCODE

---

**Algorithm 1:** PyTorch pseudocode for BaDLoss.

---
**Input**      : Training data $X$, Labels $Y$, $n_{clean}$ bona fide clean data points $X_c \in X$, Number of
                 detection epochs $n_{epochs}$, parameter $k < n_{clean}$, rejection fraction $r$
**Output**     : Clean model, Loss trajectories for all training examples, Anomaly scores
**Parameter** : Learning rate $\alpha$, Loss function $\mathcal{L}$

**Initialize**  : Anomaly scores $S[]$ array for per-example anomaly scores.
**Initialize**  : Average losses $A[]$ array for rejecting epochs with loss spikes.
**Initialize**  : Model parameters $\Theta$, Loss trajectories $T[] \leftarrow$ empty list of lists
**for** *epoch* $\leftarrow 1$ **in** $n_{epochs}$ **do**
    // Train the model for one epoch on the dataset, getting the
       average epoch loss.
    $\theta, l \leftarrow$ train_model$(X, Y, \theta, \alpha, \mathcal{L})$
    If ($l > 2$ * average(A[-3:])): continue
    $A$.append(l)
    // Collect loss trajectory at the end of the epoch
    T.append([])
    **for** *each* $(x, y)$ **in** $(X, Y)$ **do**
        predictions $\leftarrow$ model.forward(x) // no gradients are calculated
        loss $\leftarrow \mathcal{L}$(predictions, y)
        T[epoch].append(loss)
    **endfor**
**endfor**
// Calculate anomaly scores
**for** *each* $x$ **in** $X \backslash X_c$ **do**
    nearest_clean_neighbors $\leftarrow$ nearest_neighbors$(T[x], T[X_c], k)$ // get n_2 nearest
       clean trajectories
    distances $\leftarrow$ distance_metric$(x,$ nearest_clean_neighbors$)$
    $S$.append(average(distances))
**endfor**

anomaly_scores $\leftarrow$ S
// Any clean training method can be used
// Typically, removing examples below a threshold works well
model = clean_train$(X, Y, r,$ anomaly_scores$)$
**Output** model, $T$, anomaly_scores
**END**

---

## B  DIFFERENCES BETWEEN BADLOSS AND ABL

Anti-Backdoor Learning (ABL) (Li et al., 2021a) hinges on the idea that the loss of a particular example is informative about whether the example is clean or backdoored. However, ABL's detection mechanism relies on a very strong assumption regarding the learning of the backdoored instances i.e., the loss on these backdoored instances should be much lower than clean instances at the end of 20 epochs of training.

As these choices are tuned specifically for the datsets evaluated in the original work, they fail to generalize in a variety of circumstances, including our multi-attack setting. BaDLoss, on the other hand, just assumes that the trajectories generated by backdoored examples are distinct from clean examples, since backdoor triggers induce abnormal behaviors in the model.

Consequently, **BaDLoss – but not ABL – is agnostic as to whether a poisoned example produces higher or lower loss values, and agnostic to at which point in training those higher or lower values occur.** Therefore, ABL's filtering method can be seen as a special case of BaDLoss with only a single point in the trajectory, and the distance computation w.r.t. a fixed threshold for the assignment. Our experiments demonstrate that these differences from ABL are critical to the success of BaDLoss

in defending against simultaneous attacks. However, BaDLoss suffers from its own set of limitations. We provide a brief overview of these limitations in Section 6.4.

## C  SCALABILITY ANALYSIS

### C.1  MEMORY COMPLEXITY

BaDLoss requires storing a loss value for every single training example in the training dataset $\mathcal{D}$ for each of the training epochs in $n_{epochs} * r_{epochs}$, where $n_{epochs}$ is the total number of epochs for the full training run. Set $n_{pretrain\ epochs} = n_{epochs} * r_{pretrain}$. Consequently, the memory requirement of BaDLoss pretaining is $\mathcal{O}(|D| * n_{pretrain\ epochs})$. For instance, in our ImageNet training run with approximately 1.2 million training images and 15 epochs of pretraining, BaDLoss requires storing 18 million loss values, which, using 32-bit floating point numbers (not required – 16 or even 8-bit floats could suffice), is 72 MB of data. Additionally, as this data can be stored instead of held in memory during the training run, this has not proven to be a bottleneck. We expect even exceptionally large datasets such as LAION-5B (Schuhmann et al., 2022) could be feasibly analyzed by BaDLoss, as each image in the dataset will generate only a few bytes of associated data, yielding memory complexity penalties easily manageable at the corresponding large scale of training.

### C.2  TIME COMPLEXITY

BaDLoss' greatest computational cost is the required number of pretraining epochs, as iterating the dataset and backpropagating at every step is expensive (storing floats is $\mathcal{O}(|\mathcal{D}|)$ per epoch and is thus negligible). Call the cost of a training epoch with backpropagation $E$. BaDLoss's pretraining cost is then $\mathcal{O}(E * n_{epochs} * r_{epochs})$. We provide context for this value compared to each defense we compare against:

- **Neural Cleanse.** Neural Cleanse primarily requires training a mask for every class on the full dataset. Each training epoch for the mask costs approximately $E$. Let the number of classes equal $n_{classes}$ and the average number of training epochs for each mask to converse equal $n_{mask\ epochs}$. Neural Cleanse consequently costs $\mathcal{O}(E * n_{classes} * n_{mask\ epochs})$. In our experiments, we found $n_{mask\ epochs} \approx 20$, so for the 10-class datasets, this was $\sim 200 * E$, and for GTSRB, $\sim 860 * E$.

- **Activation Clustering.** Activation Clustering's analysis requires 1 forward pass through the training dataset, and so is negligible in cost compared to the required full retraining. The cost is approximately $\mathcal{O}(E * n_{epochs})$.

- **Spectral Signatures.** Similarly to Activation Clustering, requires only a forward pass for its analysis and requires full retraining. The cost is approximately $\mathcal{O}(E * n_{epochs})$

- **Frequency Analysis.** Frequency Analysis' cost depends on the size of the bona fide clean data, and the number of training epochs for the frequency classifier, typically 10. This cost is usually negligible, assuming $n_{clean} \ll |\mathcal{D}|$.

- **Anti-Backdoor Learning.** Anti-Backdoor Learning requires a static number of detection and unlearning epochs, $n_{detect}$ and $n_{unlearn}$ respectively. The computational cost is approximately $\mathcal{O}(E * (n_{detect} + n_{unlearn}))$.

- **Cognitive Distillation.** Requires a fixed number of optimization steps $n_{pattern}$ per training example to learn the logit-matching pattern mask (typically 100). Total cost is approximately $\mathcal{O}(E * n_{pattern})$.

- **Poisoned Sample Sensitivity.** Requires $n_{pretrain}$ epochs to warm-start the model, $n_{intraclass}$ epochs to differentiate class activations, and $n_{unlearn}$ for the alternating learning/unlearning phase after training (as the dataset is divided into a clean and dirty section beforehand, the learning and unlearning together cost about $E$). The computational cost is approximately $\mathcal{O}(E * (n_{pretrain} + n_{intraclass} + n_{unlearn}))$.

- **Causal Backdoor Defense.** Requires a fixed number of pretraining epochs (typically 10) which we will discount. At every training epoch, the discriminator is first trained by forward passing the base model twice, then forwards and backwards passing the discriminator. Then, the base model and discriminator are run forward one additional time, in addition to the

normal forward and backward pass of the base model. Treating the cost of a forward pass as approximately $\frac{1}{3}$ the cost of a full epoch (as the backwards pass is about twice as expensive as the forwards pass), and $D$ as the cost of a training epoch for the discriminator, the additional cost of this defense is approximately $\mathcal{O}(E * n_{epochs} + D * n_{epochs} * \frac{4}{3})$. The discriminator is typically very small (a 2-layer MLP, operating on logits instead of images), so we can also treat $D$ as negligible, and see that the cost is approximately equivalent to full retraining: $\mathcal{O}(E * n_{epochs})$.

Consequently, we conclude that the computational cost of BaDLoss's pretraining epochs is not exceptionally high compared to the suite of existing backdoor defenses we compare against.

BadLoss' analysis incurs a much lower cost. Each of the $n_{clean}$ examples's loss trajectories needs to get its distance calculated to every other example in the training dataset. The dimensionality of each vector is $n_{epochs} * r_{epochs}$, so the overall computational complexity of this analysis is $\mathcal{O}(n_{clean} * |\mathcal{D}| * n_{epochs} * r_{epochs})$. Other analysis steps have lower time complexity (e.g. detecting loss-spike epochs requires calculating the average loss of each pretraining epoch, so $\mathcal{O}(|\mathcal{D}| * n_{epochs} * r_{epochs})$).

Comparing the pretraining and the analysis costs, we see that so long as $n_{clean} \ll \frac{E}{|\mathcal{D}|}$, i.e. that the cost of a training step on a given example requires much more than a small constant times $n_{c}lean$ floating point operations, the analysis is negligible in cost.

## D  ADAPTIVE ATTACK AGAINST BaDLoss

BaDLoss assumes that any training examples with trajectories sufficiently close to the trajectories of a small set of bonafide clean examples are similarly clean. While this is empirically effective against a variety of backdoor attacks, even in the multi-attack setting, this can be partly attributed to the fact that no existing backdoor attacks have been developed to intentionally produce clean-imitating loss trajectories. An adaptive attacker with the knowledge of our attack can potentially define an attack that intentionally imitates the trajectories of clean-looking examples, hence evading detection.

One way to realize this would be to adjust the attack strength such that the loss of each backdoor sample equals the average loss of the clean samples at every epoch. An alternate and more principled realization is to define bi-level optimization, which optimizes the trigger in the outer loop and trains the model for the evaluation of the resulting trajectories in the inner loop (similar to other meta-learning formulations). However, optimizing the trigger based on this formulation is challenging.

One important point to note is that this makes the attack extremely expensive, requiring multiple training runs. Therefore, this makes the attack practically meaningless for any viable commercial application. We consider simpler techniques for adaptive attacks on BaDLoss as an interesting direction for future.

## E  MODEL MISMATCH FOR EXISTING DEFENSES IN THE MULTI-ATTACK SETTING

In this section, we will briefly describe why, for each defense considered from the literature, we should expect that the defense will not adequately generalize to the multi-attack setting.

**Neural Cleanse.** Neural Cleanse detects a backdoor if one of the per-class anomaly scores is much higher than the median. If multiple classes are attacked, there may be many classes with elevated per-class anomaly scores. Consequently, the median is likewise elevated, and as a result, many or all attacked classes are undetected. We have verified this experimentally.

**Activation Clustering.** Activation Clustering assumes that backdoored classes will have two cleanly separable clusters in the last-layer activations. However, if a class is attacked multiple times, there will be two or more clusters, and the separation will no longer be clean. Consequently, the detection method will fail to detect attacks.

**Spectral Signatures.** Spectral Signatures expects that attacked images will produce activation patterns with lower complexity (as measured by the magnitude of those patterns when projected

onto the strongest dimensions of a singular value decomposition). Similarly to Activation Clustering, when there are multiple attacks on a class, this assumption no longer can be expected to hold.

**Frequency Analysis.** We expect that Frequency Analysis would generalize well to the multi-attack setting *if and only if* the seed patterns used for the frequency classifier are accurate to the types of backdoor used. However, this assumption does not appear to hold well with the some of the attack types we tested against.

**Anti-Backdoor Learning.** As described in Section B, Anti-Backdoor Learning is dependent on attack images having a low loss at a particular epoch. It is not only the case that attack images may not have low loss at that epoch. Additionally, when multiple low-loss attacks are present, those attacks may not be *equally* low loss. Instead, one attack may occupy all the lowest 'slots' that ABL detects, causing other low loss attacks to go unnoticed.

**Cognitive Distillation.** When many backdoor patterns are present, it is important not only to distinguish that a backdoor is present, but also that no *other* backdoors are present. Consequently, the model may need to attend to many more parts of an image in order to form its conclusion, so the minimal mask that reproduces a given set of logits may be larger, making it harder to disambiguate backdoor images from clean images.

**Causal Backdoor Defense.** Similarly to ABL, some backdoors may not be learned quickly early in training. Consequently, the model must either ignore some backdoors, or incorrectly suppress some clean features learned during the training of the intentionally backdoored model.

**Poisoned Sample Sensitivity.** We have observed that the FCT-metric used in PSS is highly sensitive to particulars of the training set-up[5]. When the strong augmentations used for the FCT metric are applied to a multi-attacked model, we expect that the activation patterns generated are very different for both clean and backdoored images. Consequently, the FCT metric does not detect a substantial difference in sensitivity between clean and backdoored images.

## F    POTENTIAL CONSEQUENCES OF MULTIPLE TRIGGERS FOR A BACKDOOR ATTACK

An attacker could attempt to install multiple triggers in a single attack with a single target label, with the goal of more effectively inducing the model to produce the desired behavior. We chose not to evaluate against this setting because BaDLoss seems likely to easily defend against such attacks. Specifically, when multiple attacks are used on the same examples, we expect that the attack will be learned faster than *any* of the component attacks, since the model will at least be able to detect the strongest, most quickly learned feature, and will then be able to use the remaining features to more quickly increase its confidence in the attack. As BaDLoss already effectively defends against attacks with faster-than-clean dynamics, we expect that this attack would be particularly easy to defend against for BaDLoss. Consequently, we do not consider this case as it would not be a valuable way to test BaDLoss's robustness.

Additionally, note that the aggregate set of multiple triggers could be framed as a single, more complex trigger.

## G    MINORITY CLASS IMPACT

Minority class examples could exhibit unusual training dynamics due to being present as a small fraction of the overall dataset. We examine GTSRB, an imbalanced-class dataset, to see if there is a substantial difference in how many clean examples are mistakenly marked as backdoors, based on the class that the example is a part of. We hypothesize that classes with fewer samples will have more erratic loss trajectories (outliers) that make them more likely to be marked as backdoor examples. We examine the top 10% of classes by weight in the training set (average of 1298 samples per class) and the bottom 10% of classes in the training set (average of 140 samples per class), and check their rejection rates when run with BaDLoss in the full multi-attack setting (likely worst-case, due to maximally erratic loss trajectories). We find the following result:

---

[5]Refer to single-attack results in Section 5.3 to demonstrate how Poisoned Sample Sensitivity's performance in our setting (i.e. with a standardized model, optimizer, etc.)

- **Top 10%** Classes in this range have 25.5% of their clean examples rejected.
- **Bottom 10%** Classes in this range have 52.5% of their clean examples rejected.

It appears that outlier clean examples are more likely to be marked as false positives. However, we note that the rejection rate of the bottom 10% classes is still lower than that of backdoor examples (in GTSRB multi-attack, BaDLoss rejects 95.1% of backdoor samples), and that the overall clean accuracy of the model remains high. Future work could investigate whether including a fixed amount of samples from each class could mitigate this disproportionate harm on minority examples without reducing detection accuracy.

## H    Training Details

We use the ResNet-50 (He et al., 2015) architecture throughout all our experiments, except in ImageNet, where we use a ResNet-18. We use the AdamW optimizer with learning rate $\gamma = 1e-3$ (with a cosine-annealing learning rate schedule), weight decay $\lambda = 1e-4$, and $\beta_1, \beta_2 = 0.9, 0.999$. We train for 100 epochs on CIFAR-10 and GTSRB, and 250 epochs on Imagenette. We intentionally undertrain for 50 epochs in ImageNet for computational efficiency – further training would improve clean accuracy. In CIFAR-10, we use a batch size of 128. In GTSRB, Imagenette, and ImageNet, we use a batch size of 256. In CIFAR-10, we use a crop-and-pad (4px max) and random horizontal flip augmentation except during BaDLoss pretraining. In GTSRB, we use no augmentations. In Imagenette and ImageNet, we use random resized crop (scale=0.08-1) and random flip augmentations. In ImageNet, we first resize all images to 256x256 for dataloading efficiency, then use random resized crop (scale=0.8-1) and random flip augmentations.

We train using PyTorch (Ansel et al., 2024). Our nearest neighbors classifier uses scikit-learn (Pedregosa et al., 2011). Plots were generated with Matplotlib (Hunter, 2007).

## I    Attack and Defense Details

### I.1    Attacks Considered

We focus on attacks that exclusively impact the training dataset. While other attacks exist, our method is not designed to defend against such threat models. Fig 3 illustrates the attacks considered in this study. Note that attacks have very different poisoning ratios $p$: for example on CIFAR-10, ranging from 0.0005 (25 images) to 0.03 (1500 images).

**Image Patch.** Gu et al. (2017) use a 4-pixel checkerboard (Patch) and a 1-pixel (Single-Pix) attack, adding a pattern and changing the corresponding image's label to the target class. We use $p = 0.01$ for both attacks in CIFAR-10. $p = 0.02$ for the checkerboard patch and $p = 0.04$ for the single-pixel patch in GTSRB. $p = 0.05$ for the checkerboard patch in Imagenette, and the attack is scaled up to be 32 pixels in size to accommodate the larger image size. $p = 0.001$ in ImageNet.

**Blended Pattern.** A blended pattern attack adds a full-image trigger $t$ into an image with some fraction $\alpha$, such that the attacked image $x_{\text{attacked}} = \alpha t + (1 - \alpha)x_{\text{original}}$. We use three blending attacks: Chen et al. (2017)'s random noise blending (Blend-R) with $\alpha = 0.075, p = 0.01$, Liao et al. (2018)'s dimple pattern blending (Blend-P) with $\alpha = 0.025, p = 0.01$, Barni et al. (2019)'s sinusoid pattern blending (Sinusoid) with $\alpha = 0.075, p = 0.01$. The sinusoid attack is a clean-label baseline, so images are only selected from the target class and their labels are not changed. As GTSRB has imbalanced classes, we require that the chosen target class for the sinusoid attack has at least 1,000 images, of which the sinusoid backdoor is applied to at least 300. In Imagenette, we increase $\alpha$ to 0.15. In ImageNet, we use Imagenette's $\alpha$, but use $p = 0.0005$ for the random blending attack, $p = 0.0002$ for the pattern blending attack, and $p = 0.0001$ for the sinusoid attack. For all blended attacks, to evaluate attack success rates, we boost the magnitude of the blended pattern (Zeng et al., 2022b) by 2x.

**Learned Trigger.** Zeng et al. (2022b) optimize a trigger pattern to embed a backdoor with very few backdoor examples. We use $p = 0.0005$ in CIFAR-10, and do not use this attack in any other dataset.

**Frequency Attack.** Wang et al. (2021) inject a trigger into an image's discrete cosine transform, acting as a high-frequency trigger pattern. We use this attack with $m = 30, p = 0.01$. In Imagenette,

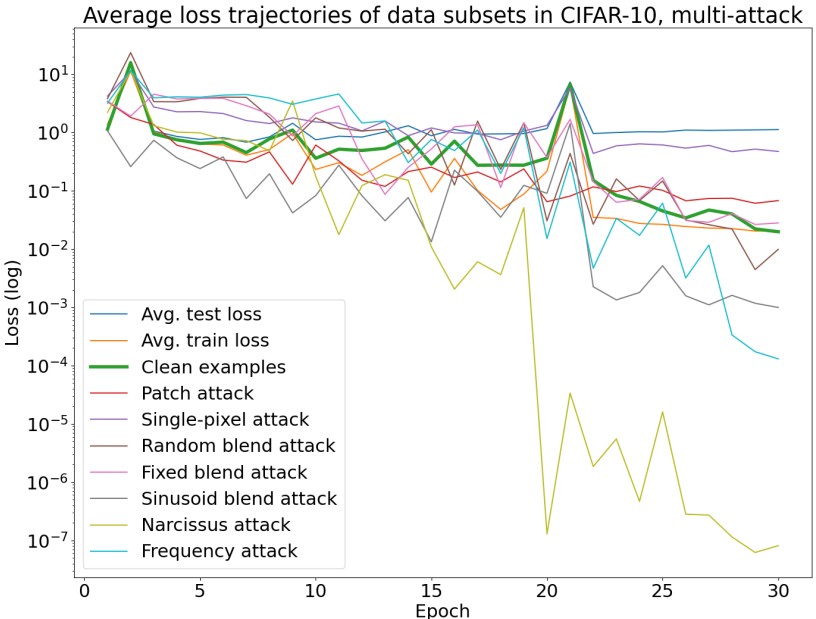

Figure 9: **Loss trajectories in the multi-attack setting in CIFAR-10.** Note loss spikes in epochs 2 and 21. Loss has been plotted on a log-scale to emphasize backdoor attacks with lower loss than clean examples. Clean examples used for BaDLoss filtering are in heavy green. Some distinct trends: The single-pixel attack is consistently above the clean examples, while the narcissus and frequency attacks start high and end up lower. Despite the patch attack examples tracking the clean examples relatively well, BaDLoss still appears capable of removing it.

we increase $m$ to 90. In ImageNet, we use Imagenette's $m$, but decrease the poisoning ratio to $p = 0.001$.

### I.1.1 META-ATTACK SETTINGS

We always choose the attack target randomly. When there are multiple attacks, we choose randomly with replacement. Target images are also chosen randomly – for clean label attacks, always from a single class. Target image choice is without replacement.

### I.2 DEFENSES CONSIDERED

We prioritize selecting defenses which are primarily filtering-based, as BaDLoss is primarily a filtering-based method. Where possible, we adapt defenses to remove and retrain from scratch for consistence.

**Neural Cleanse.** We use Wang et al. (2019)'s proposed filtering technique: finding the trigger, identifying triggered neurons, then filtering the training dataset by removing images with high trigger neuron activations, using their default threshold.

**Activation Clustering.** We use Chen et al. (2018)'s silhouette score identification method to remove poisoned data, using their default parameters.

**Spectral Signatures.** Spectral signatures always removes a fixed fraction (15%) of datapoints that are deemed most anomalous (Tran et al., 2018).

**Frequency Analysis.** We construct the frequency detector (Zeng et al., 2022c) training dataset using only the bona fide clean examples that BaDLoss has access to, for parity.

| | | Clean Acc. | Attack Success Rate | | | | | | | |
|---|---|---|---|---|---|---|---|---|---|---|
| | | | Patch | Single-Pix | Blend-R | Blend-P | Sinusoid | Narcissus | Frequency | Avg. ASR |
| CIFAR-10 | No Defense | 91.76 | 91.58 | 34.97 | 99.27 | 96.74 | 45.36 | 99.93 | 99.98 | 81.12 |
| | Neural Cleanse | 91.76 | 91.58 | 34.97 | 99.27 | 96.74 | 45.36 | 99.93 | 99.98 | 81.12 |
| | Activation Clustering | 91.83 | 92.61 | 49.24 | 99.44 | 99.21 | 44.29 | 98.04 | 99.79 | 83.23 |
| | Spectral Signatures | 90.82 | 91.50 | 3.58 | 3.31 | 99.80 | 31.48 | 94.66 | 0.20 | 46.36 |
| | Frequency Analysis | 90.96 | 94.44 | 64.38 | 97.41 | 97.70 | 73.43 | 0.00 | 99.98 | 75.33 |
| | Anti-Backdoor Learning | 90.73 | 94.00 | 71.29 | 99.14 | 99.90 | 62.86 | 4.99 | 14.92 | 63.87 |
| | Cognitive Distillation | 91.04 | 91.06 | 41.63 | 98.56 | 95.13 | 43.72 | 0.00 | 99.96 | 67.15 |
| | Causal Backdoor Defense | 70.84 | 3.54 | 2.00 | 15.79 | 6.48 | 40.11 | 31.81 | 5.69 | 15.06 |
| | Poisoned Sample Sensitivity | 89.90 | 91.00 | 44.47 | 74.53 | 99.77 | 18.37 | 99.89 | 99.93 | 75.42 |
| | BaDLoss | 88.43 | 1.71 | 0.92 | 6.16 | 4.29 | 33.52 | 7.81 | 1.44 | 7.98 |
| GTSRB | No Defense | 97.09 | 93.22 | 93.11 | 99.74 | 100.00 | 49.36 | - | 100.00 | 89.24 |
| | Neural Cleanse | 97.09 | 93.22 | 93.11 | 99.74 | 100.00 | 49.36 | - | 100.00 | 89.24 |
| | Activation Clustering | 97.82 | 94.96 | 94.32 | 97.05 | 100.00 | 46.69 | - | 100.00 | 88.84 |
| | Spectral Signatures | 96.56 | 91.91 | 91.14 | 63.60 | 100.00 | 38.08 | - | 100.00 | 80.79 |
| | Frequency Analysis | 97.43 | 94.36 | 94.17 | 0.81 | 100.00 | 54.14 | - | 100.00 | 73.91 |
| | Anti-Backdoor Learning | 97.13 | 92.83 | 93.93 | 52.93 | 100.00 | 77.52 | - | 100.00 | 86.20 |
| | Cognitive Distillation | 97.10 | 92.67 | 92.89 | 86.16 | 100.00 | 31.29 | - | 87.36 | 81.73 |
| | Causal Backdoor Defense | 71.98 | 1.82 | 5.77 | 4.26 | 31.90 | 0.17 | - | 14.05 | 9.66 |
| | Poisoned Sample Sensitivity | 96.82 | 91.88 | 91.16 | 99.64 | 100.00 | 46.74 | - | 100.00 | 88.24 |
| | BaDLoss | 94.75 | 0.13 | 0.04 | 10.32 | 0.18 | 1.75 | - | 49.32 | 10.29 |
| Imagenette | No Defense | 90.85 | 49.66 | - | 71.16 | 95.53 | 56.51 | - | 91.18 | 72.81 |
| | Neural Cleanse | 88.71 | 50.96 | - | 52.31 | 97.99 | 49.74 | - | 1.84 | 50.57 |
| | Activation Clustering | 90.22 | 49.29 | - | 47.69 | 99.86 | 62.73 | - | 0.91 | 52.10 |
| | Spectral Signatures | 89.17 | 41.04 | - | 34.85 | 19.26 | 54.91 | - | 3.03 | 30.62 |
| | Frequency Analysis | 90.85 | 49.66 | - | 71.16 | 95.53 | 56.51 | - | 91.18 | 72.81 |
| | Anti-Backdoor Learning | 88.08 | 37.24 | - | 43.00 | 4.24 | 78.39 | - | 0.99 | 32.77 |
| | Cognitive Distillation | 88.13 | 43.39 | - | 26.31 | 2.91 | 49.72 | - | 1.08 | 24.68 |
| | Causal Backdoor Defense | 58.68 | 4.51 | - | 11.23 | 8.76 | 98.33 | - | 4.68 | 25.50 |
| | Poisoned Sample Sensitivity | 87.39 | 46.06 | - | 14.85 | 84.49 | 66.19 | - | 12.71 | 44.86 |
| | BaDLoss | 85.58 | 35.56 | - | 1.67 | 3.81 | 53.18 | - | 1.62 | 19.17 |

Table 2: **Multi-attack setting: Clean accuracy and attack success rate after retraining on CIFAR-10 and GTSRB.** This table shows that the multi-attack setting is substantially harder than the single-attack setting. BaDLoss demonstrates the best overall defense in both settings, but suffers some clean accuracy degradation in CIFAR-10.

| | Clean Acc. | Patch | Blend-R | Blend-P | Sinusoid | Freq | Avg. ASR |
|---|---|---|---|---|---|---|---|
| No Defense | 57.99 | 91.20 | 99.10 | 92.45 | 68.10 | 92.00 | 88.57 |
| BaDLoss | 54.49 | 2.45 | 29.50 | 0.05 | 46.00 | 0.05 | **15.61** |

Table 3: **ImageNet multi-attack results.** BaDLoss reduces ASR of every attack, completely defending most attacks, while maintaining adequate top-1 accuracy on ImageNet.

**Anti-Backdoor Learning.** Rather than unlearn using identified examples (Li et al., 2021a), which demonstrated substantial instability during our testing, we remove the bottom 15% of examples after the pretraining phase (instead of the 1% usually removed and used for unlearning) and retrain from scratch.

**Cognitive Distillation.** Similarly to Anti-Backdoor Learning, instead of doing an unlearning phase after finding the lowest magnitude masks (Huang et al., 2023), we remove the fixed 15% of examples with the lowest magnitudes and retrain from scratch.

**Causal Backdoor Defense.** We perform CBD retraining for 100 epochs, except in Imagenette where we retrain for 250 epochs. We explored reducing the impact of the weighted cross-entropy loss to minimize clean accuracy degradation, but did not find a satisfactory setting of this that effectively traded off ASR and clean accuracy, so simply kept the original settings.

**Poisoned Sample Sensitivity** We set the number of intraclass training epochs to 3, finding that any further intraclass training epochs damages the reference model's performance too much to be useful

| | | Clean Acc. | Attack Success Rate | | | | | | | |
|---|---|---|---|---|---|---|---|---|---|---|
| | | | Patch | Single-Pix | Blend-R | Blend-P | Sinusoid | Narcissus | Frequency | Avg. ASR |
| CIFAR-10 | BaDLoss ($r = 0.05$) | 91.59 | 93.90 | 50.80 | 99.26 | 99.78 | 31.29 | 100.00 | 100.00 | 82.15 |
| | BaDLoss ($r = 0.10$) | 91.33 | 93.60 | 39.27 | 99.17 | 96.16 | 47.80 | 99.70 | 99.96 | 82.24 |
| | BaDLoss ($r = 0.20$) | 90.95 | 86.39 | 2.58 | 96.68 | 98.82 | 39.03 | 95.79 | 2.36 | 60.24 |
| | BaDLoss ($r = 0.30$) | 89.88 | 41.32 | 0.79 | 59.57 | 11.37 | 37.88 | 80.11 | 1.70 | 33.25 |
| | BaDLoss ($r = 0.40$) | 88.43 | 1.71 | 0.92 | 29.69 | 4.29 | 33.52 | 7.81 | 1.52 | 11.35 |
| | BaDLoss ($r = 0.50$) | 85.75 | 2.14 | 1.27 | 6.78 | 5.63 | 25.49 | 1.80 | 1.59 | 6.39 |
| | BaDLoss ($r = 0.60$) | 81.69 | 2.66 | 1.94 | 3.97 | 8.31 | 28.92 | 2.91 | 2.01 | 7.25 |
| | BaDLoss ($r = 0.70$) | 80.30 | 2.90 | 2.28 | 3.79 | 7.88 | 18.59 | 2.22 | 2.57 | 5.75 |
| GTSRB | BaDLoss ($r = 0.05$) | 97.55 | 78.39 | 87.86 | 58.87 | 98.75 | 41.28 | - | 93.48 | 76.44 |
| | BaDLoss ($r = 0.10$) | 96.84 | 0.06 | 0.51 | 47.92 | 92.21 | 52.80 | - | 95.48 | 48.16 |
| | BaDLoss ($r = 0.20$) | 97.59 | 0.07 | 0.27 | 32.29 | 65.91 | 23.06 | - | 94.05 | 35.94 |
| | BaDLoss ($r = 0.30$) | 96.06 | 0.15 | 0.08 | 25.51 | 10.13 | 25.29 | - | 85.66 | 24.47 |
| | BaDLoss ($r = 0.40$) | 94.75 | 0.13 | 0.04 | 10.32 | 0.21 | 3.15 | - | 49.32 | 10.53 |
| | BaDLoss ($r = 0.50$) | 91.45 | 0.20 | 0.00 | 22.83 | 0.33 | 3.00 | - | 3.38 | 4.96 |
| | BaDLoss ($r = 0.60$) | 89.49 | 0.18 | 0.45 | 3.13 | 0.37 | 0.27 | - | 1.03 | 0.91 |
| | BaDLoss ($r = 0.70$) | 87.95 | 0.36 | 0.11 | 0.52 | 0.48 | 5.68 | - | 0.66 | 1.30 |

Table 4: **Investigation of rejection threshold for BaDLoss.** The CIFAR-10 results were computed first, and based on the results, the threshold of $r = 0.4$ was chosen. A subsequent test on GTSRB reveals that $r = 0.4$ is a reasonable stopping point, though not all attacks are fully removed at this point (most notably the sinusoid attack still remains partially-present in the final model).

| | | Clean Acc. | Attack Success Rate | | | | | | | |
|---|---|---|---|---|---|---|---|---|---|---|
| | | | Patch | Single-Pix | Blend-R | Blend-P | Sinusoid | Narcissus | Frequency | Avg. ASR |
| CIFAR-10 | BaDLoss (DenseNet) | 81.40 | 1.27 | 2.14 | 1.93 | 5.44 | 28.33 | 0.64 | 0.94 | 5.81 |
| | BaDLoss (SqueezeNet) | 58.62 | 0.00 | 11.40 | 0.00 | 7.02 | 0.00 | 2.40 | 0.00 | 2.97 |
| | BaDLoss (VGG-16) | 82.81 | 4.34 | 1.03 | 33.63 | 10.93 | 38.82 | 6.61 | 1.18 | 13.79 |
| | BaDLoss (ResNet-18) | 86.03 | 1.79 | 1.94 | 3.26 | 3.34 | 17.22 | 2.06 | 1.54 | 4.45 |
| | BaDLoss (ResNet-34) | 85.35 | 4.18 | 1.04 | 77.53 | 14.62 | 15.20 | 1.17 | 1.56 | 16.47 |
| | BaDLoss (ResNet-50) | 88.43 | 1.71 | 0.92 | 29.69 | 4.29 | 33.52 | 7.81 | 1.52 | 11.35 |

Table 5: **Ablation study of BaDLoss architecture.** While other architectures achieve superior ASR performance, ResNet-50 achieves the most-competitive clean accuracy. Note that the variance of results per architecture can also be high.

for sample detection. We use the unlearning method, rather than the secure-training method which retrains from scratch.

## J  ABLATIONS

### J.1  REJECTION THRESHOLD

In Table 4, we show our ablation result for the rejection threshold $r_{reject}$, which most influences the tradeoff between maximizing clean-accuracy and minimizing attack success rate.

### J.2  ARCHITECTURE

In Table 5, we show our analysis of BaDLoss's performance on different neural network architectures. ResNets appear reasonably robust for detection and retraining. In future work, we could use a network optimized for detection for the pre-training phase (e.g. with lower compute cost to permit more pre-training epochs) and use a more performant network for final training.

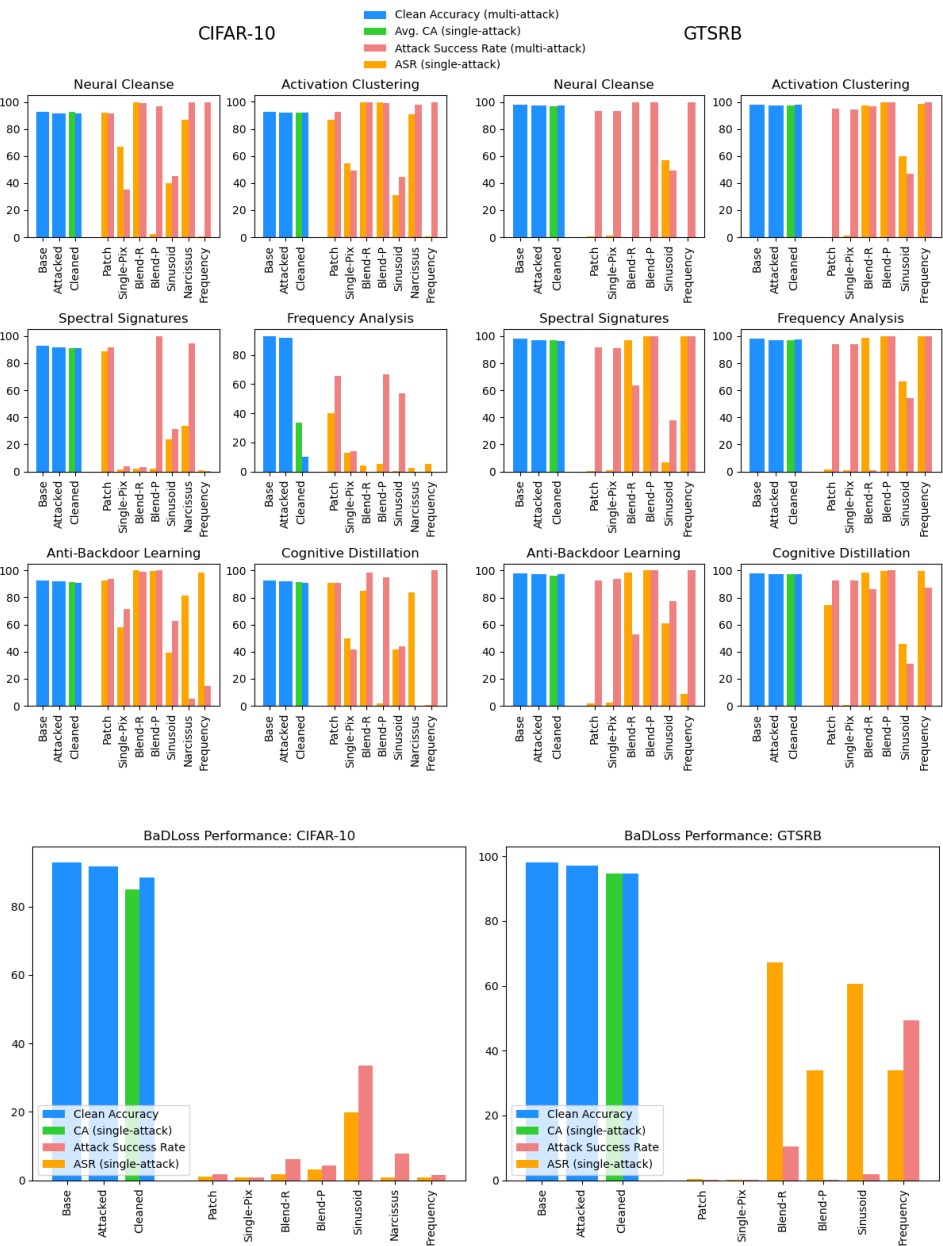

Figure 10: This figure demonstrates the difference in performance for different defenses between the single and multi-attack setting. **Top:** Existing defense performance is erratic between the single and multi-attack settings, but areas with low orange bars (attack is defended in isolation) but high red bars (attack succeeds when executed with other simultaneous attacks) are visible. **Bottom:** BaDLoss performance is highly stable regardless of single or multi-attack, and in GTSRB even improves on average in the multi-attack setting.

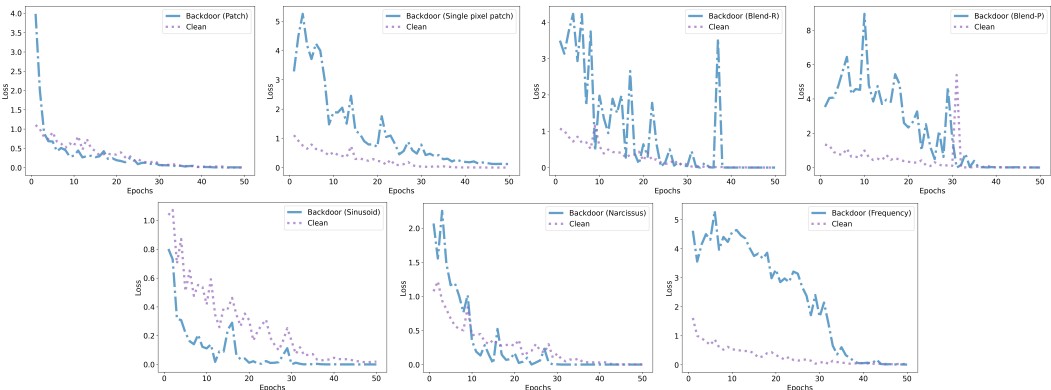

Figure 11: Average clean trajectory compared to average attack trajectories in CIFAR-10, single-attack setting, 50 epochs. **Top row, left to right:** 4-pixel patch, single-pix patch, random blended, fixed pattern blended. **Bottom row, left to right:** Sinusoid blended, narcissus, frequency attack. All backdoor attacks exhibit distinct learning dynamics from clean examples. However, some are learned faster while others are learned slower, making the inductive bias of previous methods (Li et al., 2021a; Khaddaj et al., 2023; Hayase et al., 2021) inappropriate for defending against general poisoning attacks.

## K   FULL CIFAR-10 TRAJECTORY COMPARISON

A full version of Figure 2.

