# OpenReview forum: "Protecting against simultaneous data poisoning attacks"
_ICLR.cc/2025/Conference — ICLR 2025 Poster_

### Official Review · Reviewer_jwNi · 2024-10-31

**Soundness:** 2
**Presentation:** 3
**Contribution:** 2
**Rating:** 3
**Confidence:** 4

**Summary:**

This paper proposes BaDLoss, a defense method designed to detect and mitigate multiple simultaneous backdoor attacks in machine learning models. By analyzing the loss dynamics of individual training samples, BaDLoss identifies and filters out poisoned data, effectively lowering attack success rates while preserving model accuracy. The approach is evaluated across multiple datasets and attack types, demonstrating strong performance and adaptability, making it a robust solution for complex, real-world backdoor attack scenarios.

**Strengths:**

The paper presents a novel approach to defending against simultaneous multi-backdoor attacks, addressing a challenging scenario that previous work has largely overlooked. The proposed BaDLoss method effectively utilizes loss dynamics to detect poisoned data, reducing attack success rates while preserving model accuracy. Furthermore, the paper provides a comprehensive evaluation across multiple datasets and attack types, showcasing the method's robustness and adaptability in diverse training environments.

**Weaknesses:**

1. The paper’s threat model could be clarified. If attackers control the training data, it would be helpful to understand why defenders would still need to detect and retrain on filtered data post-contamination, rather than directly identifying and removing poisoned samples before the training process.
2. The paper states that no current defenses can detect multiple backdoor samples. However, "Towards A Proactive ML Approach for Detecting Backdoor Poison Samples" presents a proactive method that addresses various backdoors. Including a comparison or discussion of this approach could provide a more comprehensive evaluation.
3. The paper does not evaluate the proposed defense method on large-scale datasets such as ImageNet. Evaluating on larger datasets would help demonstrate the method's scalability and practical effectiveness, as backdoor detection can be more challenging with increased data volume and complexity.

**Questions:**

1. How scalable is BaDLoss when applied to large-scale datasets like ImageNet, and what are the computational costs associated with tracking loss dynamics across extensive data?
2. How effective is the proposed method in defending against dynamic backdoor attacks, where attackers modify the poisoning strategy to mimic clean data loss dynamics?

---

> ### Author Response · Authors · 2024-11-28
> **Additions due to reviewer feedback (ImageNet evaluation)**
>
> We thank the reviewer for their detailed review. We appreciate that the reviewer found our work novel, comprehensive, and effective. We hope to address the reviewer’s remaining concerns below.
>
> > However, "Towards A Proactive ML Approach for Detecting Backdoor Poison Samples" presents a proactive method that addresses various backdoors. Including a comparison or discussion of this approach could provide a more comprehensive evaluation.
>
> We have not encountered this paper before, so we thank the reviewer for the pointer to this insightful work. While this work did not evaluate on the multi-attack setting, the general insight is applicable here – that the defender has more options than just blindly training in the same way as for their production-ready model. We have added a reference to this work and a brief discussion of this proactive approach, and how its insights could be adapted to our defense in Section 4.1. For example – the defender could optimize the pre-training procedure to stabilize loss trajectories and better differentiate clean and poisoned examples (e.g. by using a different architecture, batch size, optimizer than would be used for final training).
>
> > Evaluating on larger datasets would help demonstrate the method's scalability and practical effectiveness, as backdoor detection can be more challenging with increased data volume and complexity.
>
> We thank the reviewer for the suggestion. We did not previously evaluate on ImageNet due to the computational concerns – a full evaluation would require running over 50 runs to capture the full (defense x attack) space. However, at the reviewer’s request, we have evaluated BaDLoss alone on ImageNet-1k to demonstrate its scalability – similarly to previous works such as "Towards A Proactive ML Approach for Detecting Backdoor Poison Samples" mentioned by the reviewer. We provide the results in a table below. These results have also been added into the paper.
>
> | Method     | Clean Acc. | Patch | Blend-R | Blend-P | Sinusoid | Freq  | Avg. ASR |
> |------------|------------|-------|---------|---------|----------|-------|----------|
> | No Defense | 57.99      | 91.20 | 99.10   | 92.45   | 68.10    | 92.00 | 88.57    |
> | BaDLoss    | 54.49      | 2.45  | 29.50   | 0.05    | 46.00    | 0.05  | 15.61    |
>
> (The clean accuracy even without any attacks (58.54%) is lower than expected for ImageNet because we train for only 50 epochs with ResNet-18, with no optimization of the training set-up (e.g. adjusting LR schedule))
>
> As in other datasets, multiple attacks continue to be learned in concert. As suggested by the reviewer, the poisoning ratios used in ImageNet are much smaller (total poisoning ratio: 0.28%; no attack above 0.1%), so this further reinforces our claim that multiple simultaneous attacks can be effectively deployed against large, web-scale datasets. Furthermore, as in other datasets, BaDLoss induces minimal top-1 clean accuracy degradation, while decreasing ASR across the board. Though the number of training samples increases from 50K max (CIFAR-10) to 1.2M, and the number of classes increases from 43 max (GTSRB) to 1,000, BaDLoss continues to be effective with only 250 bonafide clean examples and the same set of parameters.
>
> > How scalable is BaDLoss when applied to large-scale datasets like ImageNet, and what are the computational costs associated with tracking loss dynamics across extensive data?
>
> We refer you to the table above regarding results on ImageNet. Empirically, we encountered no memory or time-complexity issues on ImageNet. Regarding the computational costs, we have added a section calculating and discussing memory complexity and time complexity in the Appendix (Section C: Scalability Analysis). We find that BaDLoss’s memory complexity is manageable, even for massive datasets such as LAION-5B, and that its time complexity is on-par with the other defenses we compare against.

---

> ### Author Response · Authors · 2024-11-28
> **Responses to reviewer questions and concerns**
>
> > If attackers control the training data, it would be helpful to understand why defenders would still need to detect and retrain on filtered data post-contamination, rather than directly identifying and removing poisoned samples before the training process.
>
> To clarify: the attackers don’t fully control the training data, but are able to poison some fraction of it. The defender doesn’t know which examples are poisoned and BaDLoss is a method that serves to identify poisoned samples, similar to previous works [1, 2, 3]. BaDLoss works by briefly pre-training on the entire unfiltered dataset for the sole purpose of identifying poisoned examples. Once they’ve been identified, they are filtered out and the model is trained from scratch on the remaining examples. We call this “retraining” simply to distinguish from the training that occurs during the identification phase.
>
> [1] Tran, B., et al. “Spectral Signatures in Backdoor Attacks.” (2018).
>
> [2] Li, Y., et al. “Anti-Backdoor Learning: Training Clean Models on Poisoned Data.” (2021).
>
> [3] Chen, W., et al. “Effective Backdoor Defense by Exploiting Sensitivity of Poisoned Samples.” (2022).
>
> > How effective is the proposed method in defending against dynamic backdoor attacks, where attackers modify the poisoning strategy to mimic clean data loss dynamics?
>
> We have discussed this in the Appendix (Section D: Adaptive Attacks Against BaDLoss). The primary difficulty with adaptive attacks is the need to back-propagate from aggregate loss statistics of a training epoch to individual images. In our experiments, this bi-level optimization problem posed substantial difficulties.

---

> > ### Comment · Reviewer_jwNi · 2024-12-03
> > **Thanks for your responses**
> >
> > Thank you for your response and the additional experiments. However, I still believe that sample detection methods, such as "Towards A Proactive ML Approach for Detecting Backdoor Poison Samples," can fundamentally eliminate various poisoned samples, effectively preventing multi-attack scenarios. Therefore, I maintain my view that directly detecting poisoned samples at the dataset level would be more effective than the existing methods.

---

> > > ### Author Response · Authors · 2024-12-04
> > > **Scope of work and clarification on dataset-level detection**
> > >
> > > We agree that "Towards A Proactive ML Approach for Detecting Backdoor Poison Samples" presents many insights useful for backdoor defense. We incorporated discussion on how BaDLoss could use the defender's 'home-field advantage' to improve detection by optimizing pre-training hyperparameters (e.g. larger batch sizes, different architectures, etc.) to increase the differences between clean and backdoor samples' losses.
> > >
> > > However, while the intersection between these two works is an interesting direction for future work, it is out-of-scope for this paper. The referenced paper's CT defense shows neither results on the multi-attack setting, nor any evidence that it would be effective in this setting. As-is, we believe our evaluations are sufficient to show that existing defenses, if not specifically tested in multi-attack, can fail dramatically in this new setting.
> > >
> > > Additionally, we do not understand the reviewer's objection that "directly detecting poisoned samples at the dataset level would be more effective than the existing methods." We note:
> > > 1. BaDLoss does directly detect poisoned samples at the dataset level, and
> > > 2. It does so by performing a pre-training and detection procedure, similar in principle to the referenced paper.
> > >
> > > In summary: we have found a problem setting that the reviewer agrees is both challenging and largely overlooked, demonstrated that a variety of existing defenses fail here, and presented an effective method for the new problem setting using dataset-level detection. We hope our work will be evaluated with this in mind.

---

### Official Review · Reviewer_GfT1 · 2024-11-01

**Soundness:** 3
**Presentation:** 3
**Contribution:** 2
**Rating:** 6
**Confidence:** 3

**Summary:**

The author considers an interesting scenario where multiple attackers simultaneously poison a dataset. The author finds that traditional poisoning methods fail in this context. Therefore, a defense method called BaDLoss, based on loss trajectories, is proposed to be robust against multiple simultaneous backdoor attacks.

**Strengths:**

1. I think simultaneously poison is an interesting topic.

2. A well-organized presentation and clear article structure

**Weaknesses:**

1. Although the method is relatively simple, I find it acceptable.

2. While the experiments are comprehensive, they lack analysis.

**Questions:**

1. Although the author provides a brief overview of why existing methods fail under simultaneous poisoning, I believe this is insufficient. A detailed theoretical explanations of why each defense fails under multiple attacks is needed.

2.The reasons for the failure of some methods are not always applicable. For example, the author mentions that Neural Cleanse fails because too many other classes are attacked, but this scenario is not guaranteed to occur. The author should provide a sensitivity analysis showing how Neural Cleanse's performance changes as the number of attacked classes increases. This would give readers a clearer understanding of when this defense starts to break down in multi-attack scenarios.

3. I'm wondering why STRIP was not included if there are specific reasons, the author should include STRIP in their experimental comparisons.

4.I'm wondering why BaDLoss performs differently across various datasets. This inconsistency in defense effectiveness between datasets requires further analysis. The authors should  provide a more detailed analysis of how dataset characteristics (e.g., number of classes, image complexity, dataset size) might influence BaDLoss's performance.

Typos:
In line 370, there is an extra "with".

---

> ### Author Response · Authors · 2024-11-28
> **Additions to paper and responses to reviewer questions**
>
> We thank the reviewer for their feedback, and we are glad that they found our work interesting and clear.
>
> > Although the method is relatively simple, I find it acceptable.
>
> We would argue that the simplicity of the method is a strength rather than a weakness. As mentioned in Section 6.1, it makes our method more flexible and applicable to a greater variety of scenarios, since our highly accurate detection method can be integrated with other parts of a backdoor defense pipeline, such as Anti-Backdoor Learning’s unlearning method.
>
> > While the experiments are comprehensive, they lack analysis.
>
> We would appreciate further clarification from the reviewer on what types of analysis they would like to see. Given the breadth of the results, it is challenging to discuss every attack/defense combination.
>
> > A detailed theoretical explanations of why each defense fails under multiple attacks is needed.
>
> We thank the reviewer for this feedback, and will be incorporating it into our paper. We have added Section E in the Appendix (“Model Mismatch For Existing Defenses in the Multi-attack Setting”) that discusses, for each defense we compare against, why the defense’s model of a backdoor may not generalize well to the multi-attack setting.
>
> > For example, the author mentions that Neural Cleanse fails because too many other classes are attacked, but this scenario is not guaranteed to occur. The author should provide a sensitivity analysis showing how Neural Cleanse's performance changes as the number of attacked classes increases.
>
> We note that running this test is highly challenging – a particular set of 2 or 3 attacks may or may not be detected by Neural Cleanse, and rigorous coverage of arbitrary combinations runs into issues with combinatorial explosion. Moreover, we have shown through our experiments in Imagenette that even 5 attacks suffice to break Neural Cleanse. Given that large-scale datasets can potentially be attacked far more than 5 times, this will only provide limited information.
>
> > I'm wondering why STRIP was not included if there are specific reasons, the author should include STRIP in their experimental comparisons.
>
> We agree that STRIP is an important baseline in the backdoor defense literature, and we include a reference to it as a key detection technique. However, we are ultimately limited in how many defenses we can compare against, as the computational cost of additional defenses grows substantially with the number of datasets and the number of attack settings used. Consequently, we hope that the reviewer will find our set of 8 defenses, both classic and modern, provides a sufficient baseline.
>
> > I'm wondering why BaDLoss performs differently across various datasets. This inconsistency in defense effectiveness between datasets requires further analysis. The authors should provide a more detailed analysis of how dataset characteristics (e.g., number of classes, image complexity, dataset size) might influence BaDLoss's performance.
>
> We thank the reviewer for this feedback. We have added a brief discussion of these differences as Section 6.2 (Differences Between Datasets). We also note that we have added results on ImageNet (Section 5.2: ImageNet Evaluation), which provides additional information regarding BaDLoss’s performance when dataset size and number of classes are varied. As these two variables appear robust, we conclude that BaDLoss’s relatively worse performance in Imagenette is due to a lower dataset size, and that BaDLoss may not be appropriate for low-data domains.
>
> > Typos: In line 370, there is an extra "with".
>
> We thank the reviewer for their attention to detail, and have corrected this issue.

---

### Official Review · Reviewer_Pi8N · 2024-11-01

**Soundness:** 2
**Presentation:** 3
**Contribution:** 3
**Rating:** 6
**Confidence:** 4

**Summary:**

The paper proposes BadLoss, a backdoor defense that removes poisoned samples by measuring the distance of samples' loss trajectories from the trajectories of a clean set. The defense is validated on poisoned training data containing multiple backdoor attacks that use patch, blend and frequency triggers. The paper compares the efficacy of the defense against a set of standard defenses and observes that existing defenses are ineffective for the examined setting.

**Strengths:**

- The writing is clear and easy to follow.
- The defense is well-motivated and the method makes sense.
- The defense requires significantly less clean data than defenses that require fine-tuning.
- Defending against simultaneous data poisoning attacks is an important and understudied topic.

**Weaknesses:**

- Studying the multi-attack setting on small benchmarks requires very high poisoning rates and results in very odd training dynamics. This is especially concerning when evaluating a defense that relies so heavily on training dynamics.

- The motivation for the analysis in section 3.3 is not clear. The classic motivation for not degrading clean accuracy is to ensure the stealthiness of the backdoor [1]. However, if the poisoning is visible from the erratic loss/accuracy curves of the model, the point is largely moot. An analysis on that would have been more valuable.

- Figure 8 provides important information on loss trajectories, but is cluttered and difficult to interpret. The legend is covering a lot of the figure. Maybe it could be split into multiple figures?

- The clean accuracy degradation and required rejection rate is quite high. As shown in table 3, the defense requires 40% rejection rate to completely remove a patch attack from CIFAR-10.

- Some phrasing in the threat model is vague, particularly: "The defender has complete control over the training process, while the attacker has neither **knowledge** nor control.'' What is knowledge of the training process? The model architecture? What classes are in the dataset? What specific attacker capabilities are being included/excluded here?

- Potentially missing references on the multi-attack setting: [2] explores the detection setting for multiple backdoors on image classifiers. [3] corroborates the patch boosting effect observed by the authors.

**Questions:**

- The above weaknesses section contains some questions. Also, were the hyperparameters of the defense tuned against the same set of attacks you evaluate against? It would be interesting to see how hyperparameters tuned to defend against a patch attack work for other types of attacks.


 - Do you expect that the unstable training dynamics would persist when training multi-attacks on a larger benchmarks where the required poisoning rate would be lower?

- In figure 8 it looks like the test loss is trending up throughout the entire training run? Could the defender have prevented successful backdoor via early stopping?

References

[1] Gu, Tianyu, et al. "Badnets: Evaluating backdooring attacks on deep neural networks." (2019).

[2] Xiang, Zhen, David J. Miller, and George Kesidis. "Post-training detection of backdoor attacks for two-class and multi-attack scenarios." (2022).

[3] Schneider, Benjamin, Nils Lukas, and Florian Kerschbaum. "Universal Backdoor Attacks." (2023).

---

> ### Author Response · Authors · 2024-11-28
> **Additions due to reviewer feedback (ImageNet evaluation)**
>
> We would like to thank the reviewer for their detailed review and valuable feedback, which we expect will improve the paper substantially. We are glad that the reviewer found our work important, novel, and logical. We hope to address the reviewer’s remaining concerns about our work here.
>
> > Studying the multi-attack setting on small benchmarks requires very high poisoning rates and results in very odd training dynamics. This is especially concerning when evaluating a defense that relies so heavily on training dynamics.
>
> > Do you expect that the unstable training dynamics would persist when training multi-attacks on a larger benchmarks where the required poisoning rate would be lower?
>
> We thank the reviewer for the suggestion to evaluate on a larger dataset that might exhibit different training dynamics. We did not previously evaluate on ImageNet due to the computational concerns – a full evaluation would require running over 50 runs to capture the full (defense x attack) space. However, to address the reviewer’s concerns, we have evaluated BaDLoss alone on ImageNet-1k to demonstrate its scalability. We provide the results in the table below. These results have also been added into the paper (Section 5.2: ImageNet Evaluation).
>
> | Method     | Clean Acc. | Patch | Blend-R | Blend-P | Sinusoid | Freq  | Avg. ASR |
> |------------|------------|-------|---------|---------|----------|-------|----------|
> | No Defense | 57.99      | 91.20 | 99.10   | 92.45   | 68.10    | 92.00 | 88.57    |
> | BaDLoss    | 54.49      | 2.45  | 29.50   | 0.05    | 46.00    | 0.05  | 15.61    |
>
> (The clean accuracy even without any attacks (58.54%) is lower than expected for ImageNet because we train for only 50 epochs with ResNet-18, with no optimization of the training set-up (e.g. adjusting LR schedule))
>
> As in other datasets, multiple attacks continue to be learned in concert. As suggested by the reviewer, the poisoning ratios used in ImageNet are much smaller (total poisoning ratio: 0.28%; no attack above 0.1%), so this further reinforces our claim that multiple simultaneous attacks can be effectively deployed against large, web-scale datasets. Furthermore, as in other datasets, BaDLoss induces minimal top-1 clean accuracy degradation, while decreasing ASR across the board. Though the number of training samples increases from 50K max (CIFAR-10) to 1.2M, and the number of classes increases from 43 max (GTSRB) to 1,000, BaDLoss continues to be effective with only 250 bonafide clean examples and the same set of parameters.
>
> > Some phrasing in the threat model is vague, particularly: "The defender has complete control over the training process, while the attacker has neither knowledge nor control.'' What is knowledge of the training process? The model architecture? What classes are in the dataset? What specific attacker capabilities are being included/excluded here?
>
> We thank the reviewer for the feedback. We have added additional details in the threat-modeling section regarding the precise knowledge the attacker has access to in Section 3.1. In short, we assume the attack has knowledge of the labels, but not of the architecture, optimizer, training algorithm, defense settings, or any other details. This is to accurately emulate the situation where the defender gets a pre-labeled dataset from the internet.
>
> > Potentially missing references on the multi-attack setting: [2] explores the detection setting for multiple backdoors on image classifiers. [3] corroborates the patch boosting effect observed by the authors.
>
> We are very extremely grateful to the reviewer for these references. We have incorporated both, as they are both highly relevant to what we have written within the paper.

---

> ### Author Response · Authors · 2024-11-28
> **Responses to reviewer questions and concerns**
>
> > The motivation for the analysis in section 3.3 is not clear. The classic motivation for not degrading clean accuracy is to ensure the stealthiness of the backdoor [1]. However, if the poisoning is visible from the erratic loss/accuracy curves of the model, the point is largely moot. An analysis on that would have been more valuable.
>
> We note that any instability in the loss/accuracy curves is not visible in aggregate – once the entire training set is averaged out, the loss/accuracy curves appear normal. These aggregate curves are what practitioners typically observe when training models. Consequently, multiple attacks remain stealthy in the conventional sense. BaDLoss’s particular insight is that erratic trajectories of individual examples can be examined to identify backdoors. In the spirit of BaDLoss, practitioners could manually examine the loss dynamics for individual training examples and make a subjective judgment about how unusual they were, but this would be extremely burdensome for large datasets.
>
> > Figure 8 provides important information on loss trajectories, but is cluttered and difficult to interpret. The legend is covering a lot of the figure. Maybe it could be split into multiple figures?
>
> > In figure 8 it looks like the test loss is trending up throughout the entire training run? Could the defender have prevented successful backdoor via early stopping?
>
> We thank the reviewer for their detailed attention to our appendices. We would like to apologize – Figure 8 was from a previous run of experiments that is not reflected in the rest of our paper. We have remade Figure 8 and included the updated version in our paper. The purpose of Figure 8 is merely to illustrate that loss dynamics are informative about whether an example is poisoned, and the new figure reflects this similarly to the old one.
>
> Early stopping would not prevent successful backdoor implantation. As previous works have shown [1, 2], certain backdoor attacks are learned very quickly in training, and thus would not be effectively prevented by early stopping. Concretely, in CIFAR-10, by epoch 10, five out of our seven evaluated backdoor attacks (all except frequency and sinusoid) are already learned to >50% ASR.
>
> [1] Li, Y., et al. “Anti-Backdoor Learning: Training Clean Models on Poisoned Data.” (2021).
>
> [2] Khaddaj, A., et al. “Rethinking Backdoor Attacks.” (2023).
>
> > The clean accuracy degradation and required rejection rate is quite high. As shown in table 3, the defense requires 40% rejection rate to completely remove a patch attack from CIFAR-10.
>
> Yes, the rejection rate is high relative to other defenses (e.g. Spectral Signatures only removes 15% of examples). We note that this higher rejection rate is due to the higher overall poisoning ratio in the multi-attack setting (due to each attack being present at the full poisoning ratio required for the attack to be learned), and a high poisoning ratio requires more rejection to ensure a clean training set. Additionally, as shown in Table 3, different attacks are different average distances from the clean trajectories, and so are removed at different ‘thresholds’. It is due to these challenges that we set our rejection threshold at 40% – but of course, this can be tuned by a practitioner to achieve a satisfactory clean accuracy while minimizing attack success rates. Note that rejecting detected backdoor examples is one possible intervention. However, our identification system is fully compatible with other methods of unlearning the detected backdoor examples, which might be more effective in unlearning in comparison to retraining, such as gradient-ascent based interventions.
>
> > Also, were the hyperparameters of the defense tuned against the same set of attacks you evaluate against?
>
> The hyperparameters of the defense were tuned on CIFAR-10 with the full suite of attacks (multi-attack), then adapted to the single-attack setting and to every other dataset.
>
> > It would be interesting to see how hyperparameters tuned to defend against a patch attack work for other types of attacks.
>
> We agree that this could be a valuable experiment, particularly because the other defenses we compared against were tuned on single-attack before we tested them on multi-attack, so this could provide an additional point of evidence regarding the efficacy of BaDLoss’s model of backdoor attacks. If we have time, we will run this experiment and provide an update here.

---

> > ### Comment · Reviewer_Pi8N · 2024-12-03
> >
> > - With the added ImageNet experiment I think its much more reasonable to consider this a defense that works in the web-scale setting.
> > - The messy figure that was confusing me has been fixed.
> > Overall, I'm leaning towards accept. I've updated my score.

---

### Official Review · Reviewer_vomn · 2024-11-03

**Soundness:** 3
**Presentation:** 3
**Contribution:** 3
**Rating:** 6
**Confidence:** 3

**Summary:**

This paper proposed a new defense, namely BaDLoss, to simultaneously defend against multiple backdoor attacks to machine learning classifiers. The proposed defense is evaluated on multiple backdoor attacks and is compared with multiple baselines on three benchmark datasets.

**Strengths:**

1. A machine learning model can be corrupted by multiple backdoor attacks. How to defend against multiple backdoor attacks simultaneously is largely unexplored in the research community.

2. The problem is well-motivated. Existing defenses are shown to be insufficient for multiple attacks, which motivates the authors to propose a new defense for this scenario. Overall, I feel the setting considered in this paper is interesting.

**Weaknesses:**

1. It is assumed that multiple attacks cannot target the same image. As acknowledged by the authors, this assumption may not hold in practice. For instance, an attacker can perform multiple backdoor attacks on the same image with the same target label.

2. The insights of the proposed defense can be discussed. For instance, the authors may consider providing insights on why backdoored samples have a different training trajectory from the clean samples.

3. Will the proposed method be effective for clean-label backdoor attacks?

4. Rejecting 40% of training samples can hurt performance in certain application scenarios, especially when the task is complex. For instance, based on Figure 6, the performance drop on Imagenette is larger than the other two datasets. The potential reason is that Imagenette is a more complex classification task.

**Questions:**

See above

---

> ### Author Response · Authors · 2024-11-28
> **Additions to paper and responses to reviewer questions**
>
> We are very grateful for this review. We are happy that the reviewer found our work to be novel and well-motivated. We hope to address the reviewer’s remaining concerns in this response.
>
> > It is assumed that multiple attacks cannot target the same image. As acknowledged by the authors, this assumption may not hold in practice. For instance, an attacker can perform multiple backdoor attacks on the same image with the same target label.
>
> We thank the reviewer for identifying this assumption and have added a short section in the Appendix (Section F: Potential Consequences of Multiple Triggers for a Backdoor Attack) to clarify why we excluded this setting. Specifically, multiple triggers on a single image would likely cause the model to learn the attack faster than any component trigger (as the model can at least attend to the most salient feature). BaDLoss is already adept at detecting fast-pattern attacks, evidenced by outperforming ABL. Consequently, we expected that BaDLoss would do particularly well in such a setting, and did not run the experiment, as we thought it would not provide much information about BaDLoss’s performance.
>
> > The insights of the proposed defense can be discussed. For instance, the authors may consider providing insights on why backdoored samples have a different training trajectory from the clean samples.
>
> We thank the reviewer for this feedback. We have added a more explicit explanation on why, based on prior work, we expect backdoored examples to develop different training dynamics in Section 4.
>
> > Will the proposed method be effective for clean-label backdoor attacks?
>
> Yes. Our method was already tested on clean-label backdoor attacks – see our results on the Narcissus and Sinusoid attack. While the Sinusoid attack is challenging to defend completely, BaDLoss still reduces the attack success rate in all datasets (including in our new evaluation on ImageNet. See Section 5.2: ImageNet Evaluation).
>
> > Rejecting 40% of training samples can hurt performance in certain application scenarios, especially when the task is complex. For instance, based on Figure 6, the performance drop on Imagenette is larger than the other two datasets. The potential reason is that Imagenette is a more complex classification task.
>
> We agree with the reviewer that the high rejection rate can harm clean accuracy. However, as we observed in Figure 7 (and in more detail in Table 4), the high rejection ratio is required in order to reduce the attack success rate of different attacks in the multi-attack setting. Naturally, the rejection ratio can be tuned by the defender to maintain a level of clean accuracy appropriate for their application. We have added further discussion in Section 6.1 on how a practitioner can adjust the rejection ratio to their needs. Since BaDLoss only provides an identification system, this can be paired with more sophisticated unlearning techniques such as gradient ascent to minimize the impact of filtering a large number of examples.

---

### Official Review · Reviewer_7Kku · 2024-11-03

**Soundness:** 2
**Presentation:** 3
**Contribution:** 3
**Rating:** 6
**Confidence:** 3

**Summary:**

This paper investigates the problem of defending against multiple simultaneous data poisoning (backdoor) attacks during the training of deep learning models and proposes a defense method called BaDLoss. Unlike previous studies, the authors point out that in real-world scenarios, models may suffer from multiple attacks simultaneously, while most existing defense methods can only handle single attacks. BaDLoss detects potential backdoor samples by tracking the loss trajectories of individual samples during training and excludes these samples from the dataset during retraining, thereby improving defense against multiple attacks without significantly degrading model performance. Through experiments on CIFAR-10, GTSRB, and Imagenette datasets, BaDLoss demonstrates superior performance compared to existing defense methods in multi-attack scenarios.

**Strengths:**

1.The paper addresses the issue of multiple simultaneous backdoor attacks, which is an important problem that has been largely overlooked in the existing literature. With the growing application of deep learning models and the complexity of training data sources, multiple attackers may tamper with the dataset simultaneously, making multi-attack scenarios a more realistic threat.

2.The paper proposes BaDLoss defense method identifies anomalous samples by tracking the loss trajectories of individual samples during training. This method fully exploits training dynamics to detect backdoored samples, regardless of whether these samples exhibit unusually high or low losses during training. Compared to traditional defense methods, BaDLoss significantly reduces the attack success rate in multi-attack scenarios while maintaining high clean data accuracy.

3.The paper validates BaDLoss on three datasets with different characteristics: CIFAR-10, GTSRB, and Imagenette, covering variations in image size, class distribution, and more. The experimental results show that BaDLoss significantly outperforms other defense methods in multi-attack scenarios.

**Weaknesses:**

1.Although the paper points out the deficiencies of existing defense methods in multi-attack scenarios, it does not provide an in-depth analysis of the specific reasons for their failures. For example, methods like Neural Cleanse and Spectral Signatures perform well in single-attack settings but fail in multi-attack scenarios.

2.The experiments in the paper assume that each image can only be affected by one type of attack, which simplifies the experimental setup. However, in real-world scenarios, a single image may contain multiple attack features, which could have a greater impact on model performance.

3.BaDLoss identifies backdoored samples by tracking the loss trajectories of all training samples, which requires storing and analyzing a large amount of intermediate data at each training stage. This could introduce additional computational overhead, especially in large-scale datasets or model applications. The authors are encouraged to provide a detailed analysis of the computational complexity of this method and discuss its feasibility and optimization strategies for practical deployment.

**Questions:**

1.In real-world applications, multiple different types of attack features may be superimposed on the same image. Can BaDLoss effectively handle such situations? Is it possible to improve the analysis of loss trajectories to further enhance the method's adaptability to complex attack scenarios?

2.In the experiments, BaDLoss outperforms other defense methods significantly under multi-attack scenarios, but its performance in single-attack settings is comparable to or slightly inferior to other methods. Have the authors considered making some adjustments to BaDLoss, such as optimizing the detection strategy for specific attack types, to improve its performance in single-attack scenarios?

3.The paper mentions that BaDLoss identifies potential backdoored samples by rejecting those farthest from clean sample loss trajectories. Could this strategy mistakenly filter out some minority class samples? In cases of imbalanced class distribution, is there a mechanism to ensure that minority class data is not mistakenly removed due to anomalous trajectories?

---

> ### Author Response · Authors · 2024-11-28
> **Additions due to reviewer feedback**
>
> We thank the reviewer for their thorough and detailed review. We agree that the problem of multiple simultaneous backdoor attacks is important and deserves attention, and we are glad that the reviewer found our results compelling. We have taken feedback into account in several areas.
>
> > Although the paper points out the deficiencies of existing defense methods in multi-attack scenarios, it does not provide an in-depth analysis of the specific reasons for their failures.
>
> We thank the reviewer for this feedback, and will be incorporating it into our paper. We have added Section E in the Appendix (“Model Mismatch For Existing Defenses in the Multi-attack Setting”) that discusses, for each defense we compare against, why the defense’s model of a backdoor may not generalize well to the multi-attack setting.
>
> > The experiments in the paper assume that each image can only be affected by one type of attack, which simplifies the experimental setup. However, in real-world scenarios, a single image may contain multiple attack features, which could have a greater impact on model performance.
>
> > In real-world applications, multiple different types of attack features may be superimposed on the same image. Can BaDLoss effectively handle such situations?
>
> We thank the reviewer for identifying this assumption and have added a short section in the Appendix (Section F: Potential Consequences of Multiple Triggers for a Backdoor Attack) to clarify why we excluded this setting. Specifically, multiple triggers on a single image would likely cause the model to learn the attack faster than any component trigger (as the model can *at least* attend to the most salient feature). BaDLoss is already adept at detecting fast-pattern attacks, evidenced by outperforming ABL. Consequently, we expected that BaDLoss would do particularly well in such a setting, and did not run the experiment, as we thought it would not provide much information about BaDLoss’s performance.
>
> > BaDLoss identifies backdoored samples by tracking the loss trajectories of all training samples, which requires storing and analyzing a large amount of intermediate data at each training stage. This could introduce additional computational overhead, especially in large-scale datasets or model applications. The authors are encouraged to provide a detailed analysis of the computational complexity of this method and discuss its feasibility and optimization strategies for practical deployment.
>
> Regarding the computational costs, we have added a section calculating and discussing memory complexity and time complexity in the Appendix (Section C: Scalability Analysis). We find that BaDLoss’s memory complexity is manageable, even for massive datasets such as LAION-5B, and that its time complexity is on-par with the other defenses we compare against.
>
> Additionally, we have briefly evaluated BaDLoss on ImageNet in the multi-attack setting and demonstrated that the computational overhead here is manageable in practice. See Section 5.2 (ImageNet Evaluation).
>
> > The paper mentions that BaDLoss identifies potential backdoored samples by rejecting those farthest from clean sample loss trajectories. Could this strategy mistakenly filter out some minority class samples? In cases of imbalanced class distribution, is there a mechanism to ensure that minority class data is not mistakenly removed due to anomalous trajectories?
>
> Yes, BaDLoss can mistakenly, and perhaps even disproportionately filter out minority class samples, as we highlighted in Section 6.4 Limitations. We agree that this is an important topic, and have added a more detailed analysis in Appendix G: Minority Class Impact. Briefly, we find that while BaDLoss does remove minority class examples more than majority class examples, the overall fraction of minority class examples removed is still far lower than the fraction of poisoned examples removed. We propose that BaDLoss’s bonafide training probes could instead be selected to ensure a minimum amount from each class to minimize the impact on minority class data.

---

> ### Author Response · Authors · 2024-11-28
> **Responses to reviewer questions**
>
> > Is it possible to improve the analysis of loss trajectories to further enhance the method's adaptability to complex attack scenarios?
>
> We certainly believe improving trajectory analysis could be a useful direction when more information about the task is available. As an example, Carlini et al. [1] used a scaled logits metric instead of cross-entropy loss to obtain distributions that are approximately Gaussian. Such alternatives could be used instead of the cross-entropy loss that we currently use to provide more informative and accurate metrics for BaDLoss detection if the application allows.
>
> [1] Carlini, N., Chien, S., Nasr, M., Song, S., Terzis, A. and Tramer, F. “‘Membership inference attacks from first principles.” (2022).
>
> > In the experiments, BaDLoss outperforms other defense methods significantly under multi-attack scenarios, but its performance in single-attack settings is comparable to or slightly inferior to other methods. Have the authors considered making some adjustments to BaDLoss, such as optimizing the detection strategy for specific attack types, to improve its performance in single-attack scenarios?
>
> We are aware that by tuning our hyperparameters for the single-attack scenario, or even re-tuning the parameters for each attack setting, we could substantially improve BaDLoss’s performance in the single-attack scenario (similar to prior work). However, we are of the opinion that this would be dishonest, as we want to present a principled defense that is robust to multiple different attacking scenarios. After all, this is our primary objection with other defenses i.e., that they fail to generalize to the multi-attack setting as they are overly tuned for that one particular setting. Over-tuning BaDLoss therefore seems counter to our goals. We are satisfied with the demonstration that BaDLoss tuned for the multi-attack setting is still competitive in single-attack.

---

### Official Review · Reviewer_UaGx · 2024-11-04

**Soundness:** 3
**Presentation:** 2
**Contribution:** 3
**Rating:** 6
**Confidence:** 2

**Summary:**

The paper addresses the challenge of defending machine learning systems against multiple simultaneous data poisoning attacks, which is increasingly relevant in real-world applications where large datasets are involved. The authors highlight the limitations of existing defenses that are effective in single-attack scenarios but fail under multi-attack conditions. The primary contributions of the paper are identification of the multi-attack threat and introduction of BaDLoss.

**Strengths:**

1. The paper presents an advancement in the field of machine learning security by addressing the novel challenge of defending against multiple simultaneous data poisoning attacks. While prior research has largely focused on single-attack scenarios, this work recognizes the complexity of real-world applications and the increased risks associated with multi-attack settings.
2. The research demonstrates high methodological quality through rigorous experimentation and robust validation of the proposed defense mechanism. The authors carefully design experiments that evaluate BaDLoss in both single-attack and multi-attack contexts, providing empirical evidence of its effectiveness.
3. The logical flow from the introduction of the problem to the presentation of BaDLoss and its evaluation ensures that readers can easily follow the authors' arguments and findings.
4. The significance of this work lies in its potential impact on the field of machine learning security. By successfully demonstrating a method that defends against multiple data poisoning attacks with minimal loss in model utility.

**Weaknesses:**

1. Limited exploration of poisoning ratio effects. While the paper acknowledges the impact of poisoning ratios on detection effectiveness, it could benefit from a more comprehensive exploration of this dimension. Specifically, the authors could conduct experiments that systematically vary poisoning ratios across different attack types to better understand the thresholds at which BaDLoss operates effectively.
2. Minority class considerations. The methodology’s tendency to mark minority-class data as anomalous could exacerbate existing imbalances in the dataset. The authors could address this issue by implementing strategies to ensure that the filtering process accounts for class representation.
3. Detailed evaluation of  counter-attacks. Although the paper briefly discusses the potential for adaptive counter-attacks, it lacks an in-depth analysis of how an informed attacker might exploit the defense mechanism. Providing case studies or simulated scenarios where attackers adapt their strategies in response to BaDLoss would enrich the discussion.
4. Limited real-world application discussion. The implications of the findings for real-world applications are somewhat underexplored. A more thorough discussion on how BaDLoss could be integrated into existing machine learning workflows or what practical challenges might arise during implementation would enhance the paper's relevance.
5. It is more reasonable to use an adaptive threshold to determine the reject ratio than a fixed ratio.

**Questions:**

1. In the context of this paper, how many levels of multiple attacks are there? Does the number of attack levels affect the performance of the model?
2. The BaDLoss proposed in this paper does not perform particularly well in a single-attack environment, with its clean accuracy being lower than that of most single-defense solutions. Does this imply that the model is only effective in scenarios involving multiple attacks?
3. Figure 2 is inconsistent with the accompanying textual description.

---

> ### Author Response · Authors · 2024-11-28
> **Additions due to reviewer feedback**
>
> We would like to thank the reviewer for their detailed analysis of our work. We agree that the problem of multiple attacks has been underexplored relative to its practical dangers, and that our experiments strongly show that this problem is both important and tractable.
>
> > Limited exploration of poisoning ratio effects. While the paper acknowledges the impact of poisoning ratios on detection effectiveness, it could benefit from a more comprehensive exploration of this dimension. Specifically, the authors could conduct experiments that systematically vary poisoning ratios across different attack types to better understand the thresholds at which BaDLoss operates effectively.
>
> We agree that this is an important consideration. We have expanded our discussion in Section 6.3 to detail some of our experiments varying the poisoning ratio in the single-attack setting. However, we note that because the poisoning ratio can be varied independently for each attack in the multi-attack setting, a fully rigorous exploration of poisoning ratios would be very computationally expensive.
>
> > Minority class considerations. The methodology’s tendency to mark minority-class data as anomalous could exacerbate existing imbalances in the dataset. The authors could address this issue by implementing strategies to ensure that the filtering process accounts for class representation.
>
> Yes, BaDLoss can mistakenly, and perhaps even disproportionately filter out minority class samples, as we highlighted in Section 6.4 Limitations. We agree that this is an important topic, and have added a more detailed analysis in Appendix G: Minority Class Impact. Briefly, we find that while BaDLoss does remove minority class examples more than majority class examples, the overall fraction of minority class examples removed is still far lower than the fraction of poisoned examples removed. We propose that BaDLoss’s bonafide training probes could instead be selected to ensure a minimum amount from each class to minimize the impact on minority class data.
>
> > Detailed evaluation of counter-attacks. Although the paper briefly discusses the potential for adaptive counter-attacks, it lacks an in-depth analysis of how an informed attacker might exploit the defense mechanism. Providing case studies or simulated scenarios where attackers adapt their strategies in response to BaDLoss would enrich the discussion.
>
> We have discussed this in the Appendix (Section D: Adaptive Attacks Against BaDLoss). The primary difficulty with adaptive attacks is the need to back-propagate from aggregate loss statistics of a training epoch to individual images. In our experiments, this bi-level optimization problem posed substantial difficulties.
>
> > Limited real-world application discussion. The implications of the findings for real-world applications are somewhat underexplored. A more thorough discussion on how BaDLoss could be integrated into existing machine learning workflows or what practical challenges might arise during implementation would enhance the paper's relevance.
>
> We appreciate the reviewer’s suggestion. We have added a brief discussion in Section 6.1 on how BaDLoss could be integrated into machine learning workflows for a practitioner.
>
> > It is more reasonable to use an adaptive threshold to determine the reject ratio than a fixed ratio.
>
> We agree with the reviewer. In general, a practitioner is able to adjust the rejection ratio for their use-case to achieve an acceptable clean accuracy while minimizing susceptibility to backdoor attacks. In the interest of simplicity, we chose a static rejection ratio that seems to perform best at backdoor removal (see Figure 7), but other methods of threshold selection are also applicable. We have added a short discussion on this topic in Section 6.1.

---

> ### Author Response · Authors · 2024-11-28
> **Responses to reviewer questions**
>
> > In the context of this paper, how many levels of multiple attacks are there? Does the number of attack levels affect the performance of the model?
>
> We are uncertain about what the reviewer means regarding “levels of multiple attack”. If the question is regarding the number of attacks – we vary between 5 attacks on Imagenette and 7 attacks on CIFAR-10, and demonstrate that in this range, the model’s clean performance is minimally affected (see Figure 4). We do not evaluate cases where there are only a small number of distinct attacks (e.g., 2-3) as we try to focus on difficult cases in this work. We assume that our defense methodology would still function well in this setting as suggested by our sustained performance between the single-attack and multi-attack settings.
>
> > The BaDLoss proposed in this paper does not perform particularly well in a single-attack environment, with its clean accuracy being lower than that of most single-defense solutions. Does this imply that the model is only effective in scenarios involving multiple attacks?
>
> We are aware that by tuning our hyperparameters for the single-attack scenario (particularly, lowering the rejection threshold), we could substantially improve BaDLoss’s performance in the single-attack scenario (similar to prior work). However, we are of the opinion that this would be dishonest, as we want to present a principled defense that is robust to multiple different attacking scenarios. After all, this is our primary objection with other defenses i.e., that they fail to generalize to the multi-attack setting as they are overly tuned for that one particular setting. Over-tuning BaDLoss therefore seems counter to our goals.
>
> Overall, we note that BaDLoss’s model of backdoors is viable in the single-attack setting – see Figure 2, which shows that loss trajectories for backdoor examples even in the single-attack setting are distinct from clean trajectories, and this seems to be corroborated by BaDLoss demonstrating the best defense in CIFAR-10 and Imagenette.
>
> > Figure 2 is inconsistent with the accompanying textual description.
>
> We thank the reviewer for their attention to detail and have adjusted the caption accordingly.

---

### Author Response · Authors · 2024-12-04
**Summary of changes from review period**

Hello all,

We would again like to thank all the reviewers for their feedback. We would like to summarize here the changes we’ve made over the review period:
* We have added results evaluating BaDLoss on ImageNet with multiple simultaneous attacks, demonstrating that our defense effectively scales to large datasets with similar performance. See Section 5.2 (ImageNet Evaluation).
* We have added Appendix C: Scalability Analysis, wherein we calculate the expected memory and time complexity of BaDLoss. We find BaDLoss’s linear memory complexity to be comfortably manageable for practical applications, and we find its time complexity to be in line with other defenses considered.
* We have added Appendix E: Model Mismatch For Existing Defenses in the Multi-attack Setting, discussing for each existing defense we evaluated against, why we might expect their implicit model of backdoor attacks to fail to generalize to the multi-attack setting.
* We have added Appendix F: Potential Consequences of Multiple Triggers for a Backdoor Attack, discussing the possibility of a backdoor attack executed with multiple triggers on a single image. In summary, we expect BaDLoss to do better on such attacks than on any of the ‘component’ triggers of the attack, as the loss trajectory will be at least as deviant as that of the strongest attack.
* We have added Appendix G: Minority Class Impact, discussing how BaDLoss affects minority class data when classes are imbalanced. In summary, minority classes are more impacted by removal than majority classes, but less so than actually poisoned data.
* We have improved the nuance and clarity of various sections, added numerous valuable references, and expanded our discussion in light of feedback from the reviewers.

We believe that, in aggregate, these changes have substantially improved our paper, and we thank the reviewers again for their contributions.

---

### Meta-Review · Area_Chair_Sh8z · 2024-12-09

**Metareview:**

The paper presents a backdoor defense method against multiple simultaneous data poisoning attacks. Reviewers appreciated its motivation and novelty. However, reviewers raised concerns about the limited exploration of hyperparameters, the risk of hurting performance, and the effectiveness against clean-label attacks. After considering the authors' response, I believe these issues can be effectively addressed through clearer explanations, illustrative examples, and additional experiments, which the authors have revised their manuscript to include. In light of the overall positive reception, I recommend accepting the paper.

**Additional Comments On Reviewer Discussion:**

Reviewer jwNi noted that directly detecting poisoned samples at the dataset level would be more effective than the existing methods, such as "Towards A Proactive ML Approach for Detecting Backdoor Poison Samples". According to the authors' response, I think they are orthogonal works. The paper's core contribution is addressing the challenge of defending against multiple simultaneous data poisoning attacks, which is a well-motivated problem.

---

### Decision · Program_Chairs · 2025-01-22

Accept (Poster)